# Recent intensified erosion and massive sediment deposition in Tibetan Plateau rivers

Jinlong Li [1], Genxu Wang[1] ✉, Chunlin Song [1] ✉, Shouqin Sun[1], Jiapei Ma[1], Ying Wang[1], Linmao Guo[1] & Dongfeng Li [2]

Recent climate change has caused an increase in warming-driven erosion and sediment transport processes on the Tibetan Plateau (TP). Yet a lack of measurements hinders our understanding of basin-scale sediment dynamics and associated spatiotemporal changes. Here, using satellite-based estimates of suspended sediment, we reconstruct the quantitative history and patterns of erosion and sediment transport in major headwater basins from 1986 to 2021. Out of 13 warming-affected headwater regions, 63% of the rivers have experienced significant increases in sediment flux. Despite such intensified erosion, we find that 30% of the total suspended sediment flux has been temporarily deposited within rivers. Our findings reveal a pronounced spatiotemporal heterogeneity within and across basins. The recurrent fluctuations in erosion-deposition patterns within river channels not only result in the underestimation of erosion magnitude but also drive continuous transformations in valley morphology, thereby endangering local ecosystems, landscape stability, and infrastructure project safety.

Rivers originating from the Tibetan Plateau (TP) not only serve as crucial water resources but also play essential ecological roles[1,2]. They transport indispensable terrigenous materials—including freshwater, sediment, carbon, and nutrients—fostering the vitality of local ecosystems and supporting the livelihoods of nearly two billion people in neighboring downstream regions. Yet, these river systems, uniquely positioned at the world's highest elevation within the mid-latitude zone, are now confronted with intensified threats from amplified climate change, rapid degradation of the cryosphere, and emerging human activities[2,3]. With the TP standing as Earth's paramount yet fragile water tower[4], it experiences a warming trend of 0.32 °C per decade[3,5], which is twice the global average. Such a pronounced warming rate, when juxtaposed with the plateau's notable altitudinal range, accentuates the susceptibility of these rivers to intricate physical and ecological processes, such as increased warming-driven erosion[6–8], deteriorating water quality[9,10], and escalating risks of natural hazards[3,11].

Existing studies highlight a substantial increase in water and sediment fluxes in TP rivers, with consequential effects on regional ecosystems and biogeochemical cycles[1,12]. For example, elevated sediment transport can curtail water clarity, potentially suppressing aquatic photosynthesis and regional primary production[13]. These sediments may also convey bioavailable micronutrients, influencing phytoplankton proliferation, and habitat suitability[14]. Notably, shifting sediment dynamics can reshape terrestrial landscapes[15], driving riverine reconfigurations[16], and destabilizing landscapes throughout the region[17,18]. Whereas riverine suspended sediment assessments are pivotal, reconstructions specific to the TP and its surrounding regions are notably absent due to insufficient high-resolution monitoring[18], even as indications underscore their markedly elevated sediment flux

[1]State Key Laboratory of Hydraulics and Mountain River Engineering, College of Water Resource and Hydropower, Sichuan University, Chengdu 610065, China. [2]Key Laboratory for Water and Sediment Sciences, Ministry of Education, College of Environmental Sciences and Engineering, Peking University, Beijing 100871, China. ✉e-mail: wanggx@scu.edu.cn; songchunlin@scu.edu.cn

relative to other cryospheric regions[1,17]. For example, between 1995-2015, TP rivers discharged sediment at rates roughly 1.8-fold[13,19] and 4.5-fold[20,21] than Greenland and pan-Arctic rivers, respectively. Furthermore, sediment transport model development is limited due to a lack of comprehensive and publicly high-quality available data[22,23]. For instance, there are limited long-term hydrological stations, with less than ~30% of TP rivers being consistently monitored (Supplementary Table 1)[1,12]. These in situ records primarily exist at discrete locations of the headwater outlets, suggesting that the warming-driven sediment dynamics within the basins are still unknown. In addition, records from hydrological stations cannot fully reflect the actual basin-scale sediment source-to-sink processes[24]. Therefore, there is an urgent need for a comprehensive spatiotemporal view of changes in sediment transport processes on the TP.

Recent advancements in remote sensing techniques[13,25,26], data availability[27], and computational resources[28] have facilitated the use of satellite imagery to enhance our understanding of sediment dynamics in ungauged headwaters. Here, we combined remote sensing and in situ river monitoring to quantitatively reconstruct the history of suspended sediment source-to-sink processes in major headwater regions on the TP. We focused on mainstreams and major tributaries of both the large rivers of Asia (e.g., the Yangtze, Yellow, Ganges, Indus, Salween, and Mekong Rivers) and inland rivers (e.g., the Amu Darya and Tarim Rivers), with an average drainage area of $>1.9 \times 10^4$ km². We utilized ~76,000 cloud-free Landsat images and billions of purely water river pixels to assess 36-year changes to fluvial suspended sediment concentration (SSC) and flux (Methods). We employed a suite of algorithms that are trained by the most accurate and comprehensive in situ datasets available[1,17]. We revealed a strong west-east spatiotemporal heterogeneity in basin-scale sediment yield and transport patterns, which is primarily driven by the combined effects of climate change and specific geomorphic processes (e.g., glacial erosion or collapses, permafrost thaw and associated landslides, rock-ice avalanches). All our findings are based on state-of-the-art advances in techniques and satellite remote sensing approaches, which can provide valuable insights into sediment mobilization, transport, and delivery in cold regions.

## Results and discussion
### Spatial heterogeneity in SSC and suspended sediment flux
We found strong spatial heterogeneity in SSC and sediment yield patterns (Fig. 1), which is gradually enlarged with the differences in climate change, local soil texture, and hydrological conditions. Nearly four-decade satellite records suggested that there was a marked decline in SSCs from northwest to southeast, with an area-weighted-average difference of $1.78 \pm 0.27$ kg/m³ (Fig. 1a; mean ± SE). However, the spatial variations of SSCs were not consistent with runoff (Fig. 1b), in contrast to the global discharge pattern of suspended sediment in river outlets (where rivers flow into the oceans and major inland seas)[18]. Satellite-estimated sediment yields also revealed significant spatial variations across the headwater basins of the TP, as well as within individual river channels. For instance, in the TP-west basins (Westerlies area), the basin-weighted sediment yield showed an annual average ranging from $229.83 \pm 56.01$ t/km²/yr in the Amu Darya basin to $891.69 \pm 125.86$ t/km²/yr in the Indus River basin. In contrast, the eastern basins of the TP (East-Asia monsoon area) displayed a smaller magnitude, with the annual average sediment yield varying from $30.67 \pm 12.98$ t/km²/yr in the Yellow River basin to $137.55 \pm 29.14$ t/km²/yr in the Salween River basin (Supplementary Tables 2 and 3). The pronounced spatial variability in sediment yield stems from the varying response levels of distinct river systems to climate change, particularly evident in the unique warming and weathering rates[28], erosion dynamics[29], and the magnitude of changes in runoff[1,2].

Temporal trends in SSCs across the TP rivers have exhibited higher increasing rates ($6.02 \pm 2.85$ %/10 yr; mean ± SE) compared to

runoff ($5.53 \pm 0.51$ %/10 yr) from 1986 to 2021 and have again revealed distinct spatial variations within each river. In highly glacierized basins in the western Himalayas, the increase rate of upstream SSCs was significantly higher ($4.53 \pm 3.26$ %/10 yr) than downstream ($1.53 \pm 0.48$ %/10 yr), despite the increasing rate of upstream runoff ($4.93 \pm 3.10$ %/10 yr) being lower than downstream ($6.05 \pm 3.25$ %/10 yr). In contrast, in the East-Asia monsoon basins, most rivers have retained a semblance of their natural variability in SSC in response to runoff. In line with changes in runoff, upstream SSCs demonstrate higher increasing rates ($1.50 \pm 0.67$ %/10 yr) than downstream SSCs ($0.63 \pm 0.23$ %/10 yr), and the timing of peak SSC remains regionally consistent with that of peak runoff. The observed spatiotemporal discrepancies between SSCs and their association with runoff suggest that hydraulic factors may not be the dominant driving force of erosion in the western region. Instead, other factors that are closely linked to the underlying surface characteristics responses to climate change, such as glacial erosion, freeze-thaw erosion of the active layer, and the pronounced influence of westerly winds during the period of low vegetation cover, may exert a more significant influence (Fig. 1). Conversely, in most areas of the eastern and southern TP, runoff continues to play an external driving role in shaping the sediment yield pattern.

High-concentration sediment export hotspots were concentrated in headwater basins dominated by glacier and permafrost ($>0.5$ kg/m³, Fig. 1). The glacierization (percentage of glacier cover per headwater basin) and snowmelt runoff ratios of six exorheic rivers were evaluated, revealing a positive correlation with the average SSCs ($P < 0.05$, Supplementary Fig. 2). At a global scale, increasing sediment yield has been the key signal of cryosphere degradation since the 1960s[12,27], primarily because of climate change-driven landscape changes, such as glacier retreat, permafrost thaw, and snowpack reduction. Such landscape disturbances have altered the magnitude and frequency of glacial erosion and thermokarst dynamics[17], resulting in increased heterogeneity in sediment yield patterns. For example, the headwater basins with the highest SSCs were the Indus basin (27.61% glacier cover) and Tarim basin (20.61% glacier cover), which are highly glaciated, with the annual average SSCs of $6.80 \pm 0.35$ kg/m³ and $4.06 \pm 1.24$ kg/m³, respectively (Fig. 1). These values were an order of magnitude greater than those of other ice-free erodible landscapes[30].

Moreover, permafrost thaw has led to the expansion of thermokarst landscapes (e.g., thaw slump, thermokarst landslide, and thaw subsidence) and escalating sediment discharge[7,12], which resulted in relatively higher SSCs in satellite imagery across permafrost regions[17,31] (permafrost cover $>50$%, Fig. 1 and Supplementary Figs. 7 and 8). At the hillslope scale, the heterogeneity of SSCs was further amplified by the interplay between permafrost soil properties, vegetation cover, and land cover characteristics[32]. Specifically, basins with higher soil bulk density and sand content are prone to significant water erosion during peak rainfall events or peak meltwater events, leading to pronounced spatial gradients in SSCs (Supplementary Fig. 3). In addition, vegetation development in proglacial and permafrost thawing areas can stabilize slopes. However, this process is complicated by deglaciation and thawing-triggered slope instability, as well as the intensification of erosive rainfall[29,33]. At the basin scale, the interplay between sediment yield and annual precipitation followed an inverted parabolic curve, with the minimum sediment yield observed at intermediate levels of annual precipitation (400–600 mm), and an increase on both sides of this minimum (Supplementary Fig. 4). With the exception of the arid Qaidam and Inner basins, areas with sparse vegetation and low ground coverage are strongly associated with high-intensity glacier and permafrost regions, where basin-scale sediment yields appear to the highest. Notably, as influences of glacier and permafrost diminish and vegetation coverage increases, sediment yield decreases accordingly. In contrast, in the eastern areas with shrub and forest vegetation, water

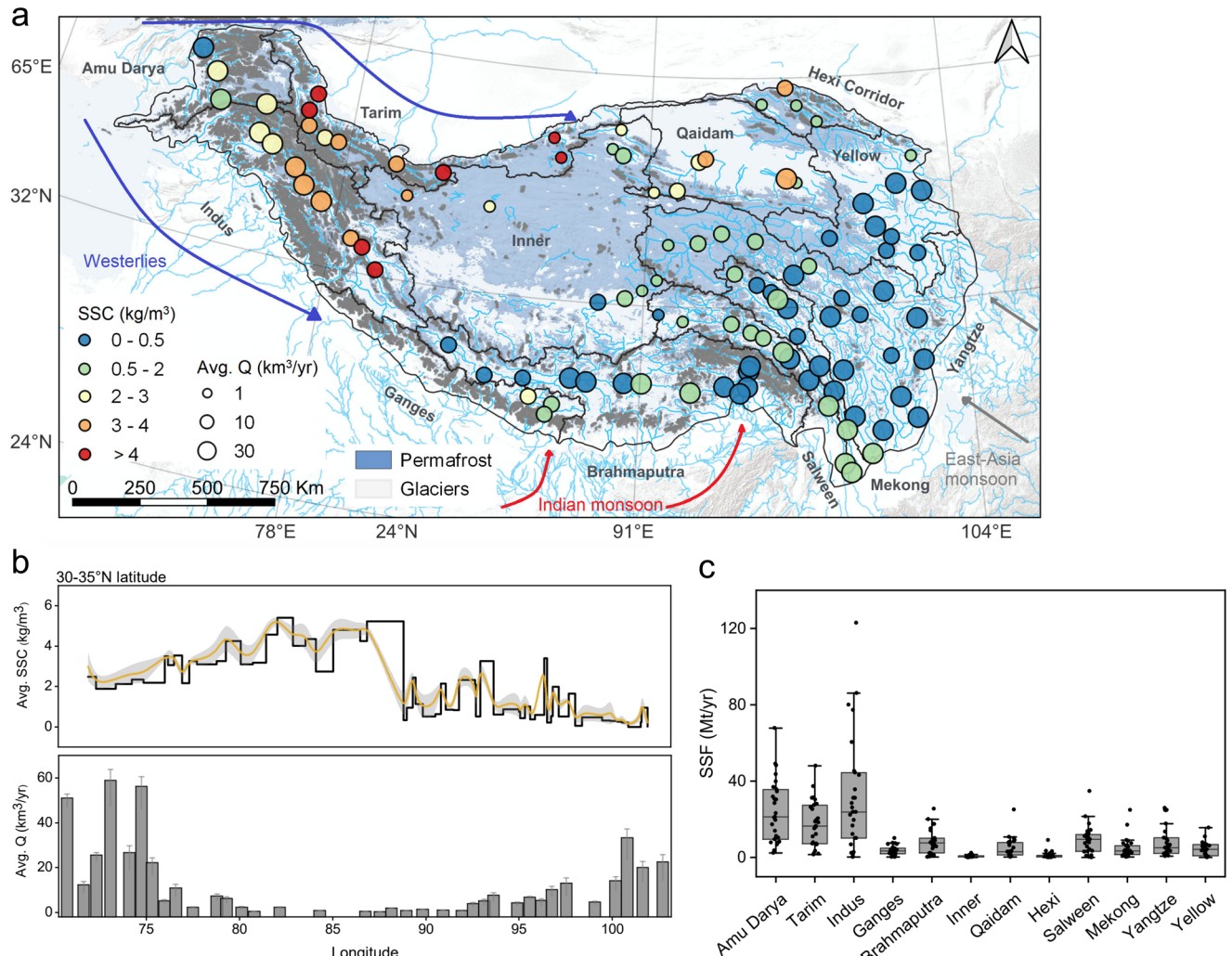

**Fig. 1 | Variations in the annual average suspended sediment concentration (SSC) and flux (SSF), as estimated from Landsat satellite images from 1986 to 2021. a** An overall map of the annual average SSCs on the Tibetan Plateau, with symbols scaled by the annual average runoff. Boundaries of glaciers and permafrost are based on refs. [67,72,75]. Base map and inset courtesy of ESRI, USGS, and NOAA. The dark-gray boundary marks the major headwater basins on the Tibetan Plateau. **b** Longitude variations in SSCs and runoff (Q) trends between 30-35°N latitude, showing a west-east divergence, with a higher average SSC and runoff values in the west and lower values in the east. **c** Variations of annual average SSF in major rivers of twelve headwater basins, based on annual satellite-estimated sediment flux statistics, with the trend being consistent with SSC. The gray-shaded area in (**b**) denotes the 95% confidence interval of the best-fit line. The black dots in (**c**) correspond to different rivers, and the horizontal lines denoted the median value of the area-weighted-average SSF for each headwater basin.

erosion driven by rainfall becomes the primary factor of sediment mobilization, with erosion rates increasing in tandem with higher rainfall intensity.

## Temporal changes in suspended sediment flux

Over the past six decades, climate change has indisputably caused significant increases in water and sediment fluxes on the TP[1,12]. At the basin scale, the satellite record spanning four decades revealed a significant increase in suspended sediment flux (SSF) in 63% of the river sections (*P* < 0.05, Fig. 2). However, this overall increasing trend masks substantial spatiotemporal variations within individual headwater basins due to limited river sampling and inadequate in situ observations[17,18]. Compared to previous studies[1,12,34], we offer several orders of magnitude more data to elucidate the pronounced spatiotemporal heterogeneity in sediment fluxes, predominantly evidenced by alterations in sediment transport dynamics.

The sediment fluxes in the western Himalayas have increased substantially in recent years, particularly in the Indus (17.87 ± 7.41 %/10 yr; mean ± SE), Amu Darya (24.57 ± 19.32% %/10 yr), and Ganges

(11.27 ± 6.42 %/10 yr) basins, due to the rapid retreat of glaciers[35] (Supplementary Table 3). Glacial erosion can mobilize and transport large amounts of sediment downstream (Fig. 2a), but the transport efficacy hinges on the connectivity between glaciers and river channels, as well as the glacier melt volume[17,36]. Satellite observations revealed that 20.98 ± 9.59% of the glacial sediment (*P* < 0.05) remains in the proglacial zone as moraines, debris cones, and alluvial fans, rather than being rapidly transported downstream (Supplementary Fig. 5 and Supplementary Table 3), resulting in a transport-limited environment. For instance, in the upper Indus basin, 41.67 ± 22.87% of annual sediment supply (reported by Darband station[34]) is temporarily stored in the proglacial zone, particularly in glacial lakes, floodplain, alluvial fans, and wider valleys that have a relatively lower terrain ruggedness indices than downstream areas (Supplementary Figs. 6 and 7). Moreover, sediment deposited near the glacier terminus and subglacial channel may be transported downstream during intermittent ice and snowmelt floods or extreme rainstorm events. Such potential spatiotemporal variations further increase the uncertainty of sediment transport dynamics, resulting in a non-significant decadal

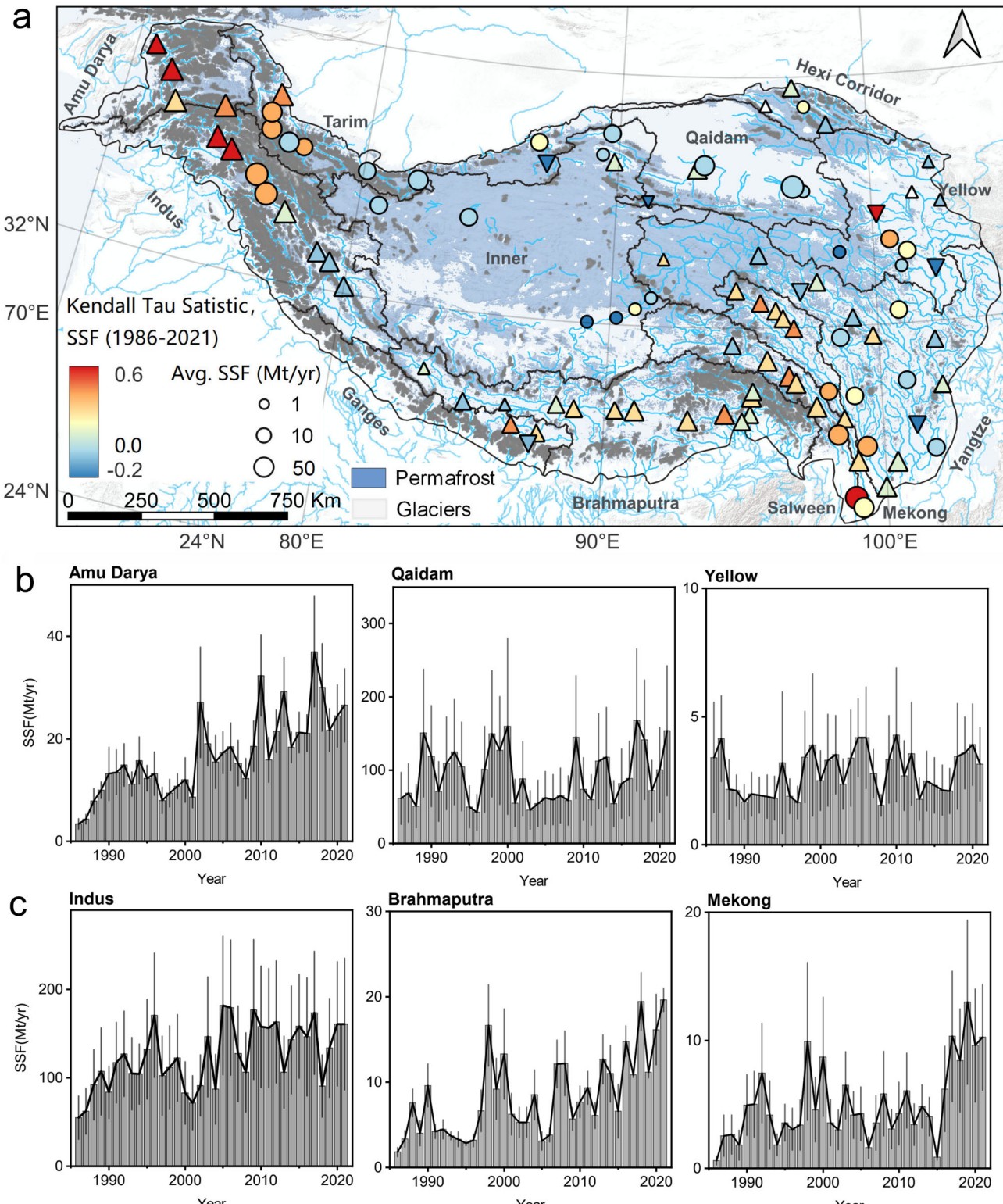

**Fig. 2 | Changes in the annual average suspended sediment flux (SSF) in 1986–2021 across the Tibetan Plateau. a** Changes in SSF for each river and the temporal trends are determined using the Mann–Kendall nonparametric trend test. The upward-pointing triangles indicate rivers with increasing SSF, while the downward-pointing triangles indicate rivers with decreasing SSF. Circles show rivers with no statistically significant change. Symbols are scaled by average SSF. Boundaries of glaciers and permafrost are based on refs. 67,75. The dark-gray boundary marks the major headwater basins on the Tibetan Plateau. Base map and inset courtesy of ESRI, USGS, and NOAA. **b** Change in temporal trends in SSF for the northern rivers (>34°N) from west to east. It reveals that SSF in the eastern regions does not change significantly but increases significantly in the western areas. **c** Change of temporal trends in SSF for the southern rivers (<34°N) from west to east. Vertical bars in (**b**) and (**c**) show standard errors. Note the axis scales vary among panels.

trend in the midstream areas of glacierized basins, in contrast to the significant increase observed upstream and downstream (Fig. 2).

In response to a warming climate, permafrost degradation and thermokarst activities have also increased sediment availability and transport capacity in the eastern permafrost headwaters. The annual increase rate in basin-weighted SSF for the Brahmaputra, Salween, Yangtze, and Mekong basins are $9.46 \pm 5.42$ %/10 yr, $18.36 \pm 8.23$ %/10 yr, $15.21 \pm 6.83$ %/10 yr, and $6.89 \pm 3.71$ %/10 yr (Supplementary Table 3), respectively, which can largely attribute to the expansion of new flow paths and active erodible landscapes[22,36,37]. The creation of new gullies and the expansion of flow paths during extreme rainstorm events have increased sediment conveyance by enhancing hillslope-channel coupling, reworking previously-stored sediment, and linking disconnected sediment sources, leading to a steady increase in sediment upstream for decades (Fig. 2 and Supplementary Figs. 5, 17). However, the signal generated by upstream permafrost thaw can be disconnected from downstream areas and the basin outlets by local sediment sinks, including expanded areas of thaw subsidence, thaw gullies, and thermokarst lakes. In arid and cold regions that comprise the upstream Yangtze, Inner ($0.41 \pm 0.19$ %/10 yr), Yellow ($4.35 \pm 3.51$ %/10 yr), and Hexi Corridor ($1.24 \pm 0.67$ %/10 yr) basins, the smaller thermokarst lakes (less than $1000$ m$^2$) account for 24% of the total TP, while larger ones (greater than $150,000$ m$^2$) account for 50%, which potentially act as important sediment sinks, effectively offsetting the downstream transport of sediment flux (Supplementary Figs. 5 and 7). Additionally, the heterogeneity of sediment transport in permafrost basins is further enhanced by changing soil permeability (Supplementary Fig. 3)[38], surface/subsurface flow paths[39], and hydrogeomorphic connectivity[37].

## Changes in riverine erosion and deposition pattern

As demonstrated in the previous studies[29,40], the changes observed in fluvial sediments in cold regions result from a combination of multiple climate change and cryosphere degradation-driven geomorphic processes. Drawing on extensive satellite data, we observed continual annual changes in riverine erosion and deposition patterns within the TP basins. Such shifts intensify the spatiotemporal heterogeneity in sediment dynamics. To gain a comprehensive understanding of the changes and to quantify the regime shift, we calculated the sediment-flux budget along the major rivers.

Compared to existing reports based on sediment monitoring stations at basin outlets[1,8,12,34,41,42], we found that the overall sediment flux of each major river has been significantly underestimated ($P < 0.05$). We conducted stringent screening to ensure data integrity, with most of our data originating from robust in situ measurements employing depth-integrated field measurement methods (Supplementary Table 2). While these data may not entirely encompass the sediment discharge trends in the large rivers, their extensive spatial coverage and temporal range enable meaningful comparisons with satellite-calibrated datasets. Specifically, the underestimation of SSC ranged from 7.81% to 43.52%, while the suspended sediment flux was underestimated between 9.01% and 46.85% (Supplementary Table 2). This is particularly true for large basins that rely solely on a single station at their outlets to represent the entire basin's erosion amount (e.g., Xiangda station in Mekong basin and Daojieba station in Salween basin, Supplementary Fig. 10). The underestimation can primarily be attributed to the dynamic redistribution of sediment within each headwater basin, due to the occurrences of sediment deposition or a lack of transport capability at specific locations, which is not adequately captured by the available station data.

We found that sediment deposition occurred in most basins (nearly >90%, Fig. 2), and we categorized these sedimentary landscapes into three types and provided a notable catalog of locations that have experienced or have the potential to experience sediment deposition (Supplementary Fig. 8). The first type of sedimentary response occurred in areas where the runoff driving force was insufficient to exceed topographic and transport thresholds, primarily in the upper and middle reaches of glacierized basins around the Himalayas, including the Indus, Tarim, Ganges, and upper Brahmaputra basins. These basins are characterized by high rock uplift, aridity, high relief, and large variability in river runoff and sediment transport[43], which have led to the development of numerous large-scale structural sedimentary basins along the TP margin, such as the Thakkhola–Mustang Graben in Nepal and the Zhada basin in the upper Sutlej region[44]. Satellite-estimated sediment budget showed that ~51.86 Mt/yr sediment has been deposited in these basins, accounting for ~18.21% of the headwater outlets export (Supplementary Table 3). More importantly, once extreme rainstorms or meltwater floods occur, these temporarily stored sediment could be remobilized and flushed downstream, causing rapid changes in regional erosion-deposition modes and significant cryosphere hazards (e.g., glacial/landslide lake outburst floods, rock–ice avalanches, rockfalls, and debris flow). For example, in 2016, a glacial lake outburst floods (GLOF) event in the Bhotekoshi/Sunkoshi River (Ganges basin, Nepal) triggered a 30-fold increase in local sediment flux through channel-connected landslides[45].

Furthermore, the second type is primarily controlled by geomorphic systems and associated hydrological connectivity, resulting in the formation of macro-distributed braided river systems and their floodplains. These sedimentary responses tend to concentrate in valley basins with steep profiles, such as the Indus, upper and middle Brahmaputra, lower Salween, and midstream and downstream areas of the Yangtze and Yellow basins. These basins are characterized by narrow gorges that alternate with wide alluvial reaches, extensive river terraces, and remnants of large lakes that are either perched or located close to the rivers (Supplementary Figs. 5 and 7). As sediment is transported to these specific locations, the river channel slope decreases, resulting in reduced sediment carrying capacity and likely deposition. This process leads to the widening and shallowing of the river channel and the accumulation of sediment onto sandbars (Fig. 3b and Supplementary Fig. 7). For instance, in the past few decades, ~21.35 Mt of total sediment (representing ~21.34% of sediment discharged downstream) has been deposited in the braided rivers and wide-narrow valley nodes of the middle reaches of the upper Yangtze River (Fig. 3c). Meanwhile, in the wide valleys of the midstream Brahmaputra River, ~45.29% of the annual sediment supply has been deposited (Fig. 3d). During the summer flood seasons, increased sediment deposition worsens the rise in the riverbed, further constraining the water cross-section and ultimately hindering sediment transport and flood discharge capacity (Supplementary Fig. 9). Our estimates indicate that over the past few decades, ~26.34 Mt/yr of sediment has been deposited at these specific geomorphic units, representing ~34.33% of the total sediment transport downstream (Supplementary Table 3).

Vegetation development can also play a role in hindering sediment transport and the expansion of erodible landscapes. In the warming TP, the overall greening and vegetation restoration can stabilize soil, reduce erosion, intercept and slow down water flow, and contribute to the formation, maintenance, and stabilization of deposition, thereby slowing erosion rates and maintaining slope stability and ecological balance[12] (Supplementary Fig. 4). However, emerging human activities such as over-exploitation, overgrazing, damming, and deforestation in some local areas can cause shifts in erosion-deposition patterns in alpine mountainous areas (Supplementary Fig. 10), exacerbating soil erosion and deposition in rivers[26,40]. Despite the significance of changes in vegetation cover, we are unable to directly detect their contribution to changes in the sediment budget. This is particularly true given that they are highly coupled with other factors; for instance, climate change, water-sediment connectivity, and related rapid adjustments in underlying surfaces, which

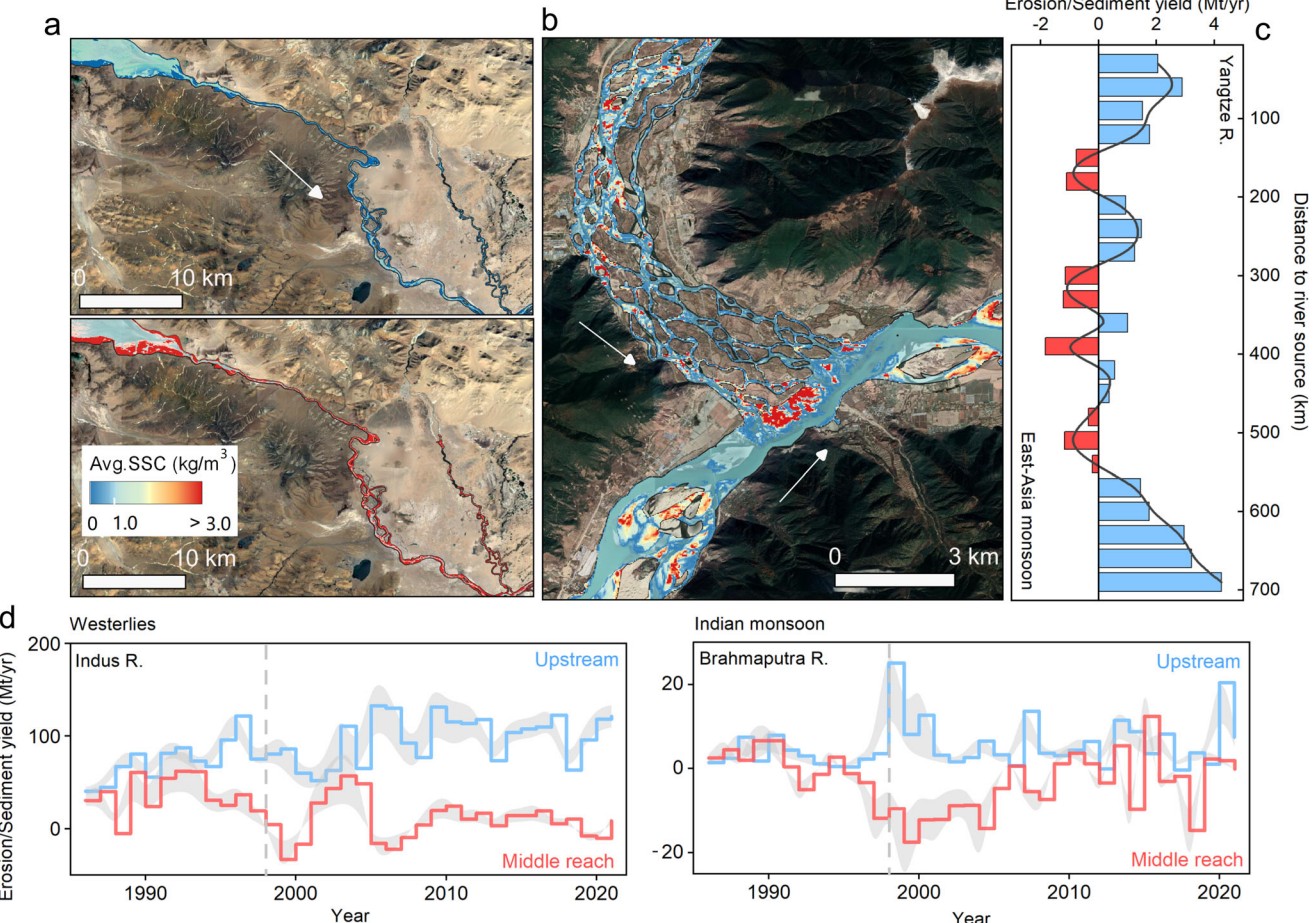

**Fig. 3 | Changes in regional erosion and deposition patterns. a** Satellite composite imagery illustrates the impacts of climate change on sediment dynamics in the proglacial rivers that originated from the Angsi Glacier, the source of the Brahmaputra River. The comparison of the upper and lower panels shows increasing suspended sediment concentrations (SSCs) from 1986–1995 (upper panel) and 2000–2016 (lower panel). **b** Satellite composite imagery (2000–2021) shows high concentrations of sediment accumulation in the braided river channels, mid-channel islands, and sandbars in the middle Brahmaputra River. These high-concentration sediments converge at the mouth of braided river channels where they meet the mainstream, resulting in extensive sediment accumulation instead of downstream transport. Base map and inset courtesy of ESRI, USGS, and NOAA. **c** Shifts of erosion/deposition patterns along the flow of the Yangtze River. The red columns are mostly distributed in braided river systems and valleys with alternating river widths. **d** Comparison of erosion/deposition modes between the upstream and midstream of the Indus River and the Brahmaputra River. Different fluvial morphologies and geomorphic systems determine significant contrasts in sediment erosion and deposition between the upstream and midstream. The gray vertical line indicates around 1998, which is typically considered a change-point year of climate change on the Tibetan Plateau[7,8] when sediment erosion and deposition in the upper and middle reaches began to change more dramatically and rapidly. The values below 0 in (**c**) and (**d**) indicate sediment deposition. The gray-shaded area denotes the 95% confidence interval of the best-fit line. Note the axis scales vary among panels.

exert more important controls on basin-scale sediment delivery processes. Overall, a series of changes in the cryospheric landscape and associated hydrogeomorphic processes can explain the observed increase and higher spatiotemporal heterogeneity in sediment transport patterns on the TP. However, it is important to note that effective sediment export still depends on regional erosion and deposition patterns.

The operation of Earth observation satellites facilitates the high-resolution mapping of suspended sediment concentrations and fluxes in remote ungauged headwater basins. Here, we used long-term Landsat data to estimate sediment transport on the TP, to quantify and capture the complex, interconnected sediment erosion/deposition patterns induced by geomorphic changes associated with global warming. We found strong spatiotemporal heterogeneity in SSC and sediment flux patterns, which is gradually enlarged with the difference in magnitude and frequency of regional landscape changes in response to climate change. Due to enhanced glacier retreat, snowpack melting, and permafrost disturbances, high amounts of sediments were

produced and transported into river channels, while differences in geomorphic patterns for each headwater basin determined sediment transport vary spatially to a large extent. ~30% of the total suspended sediment flux has been temporarily deposited in sedimentary basins, proglacial areas, and river systems with limited sediment-water connectivity or transport capacity. Once reactivated and remobilized, these previously deposited sediments can be remobilized and constitute a significant threat to the local ecology and the development of hydropower projects.

The fluvial sediment erosion and deposition we focused on were by no means confined to the TP. In fact, sedimentary and geomorphic climate change responses are expected to accelerate worldwide, especially in high mountain areas and polar regions[1,17,18,40]. Permafrost and glaciers are degrading rapidly in the high latitude polar and sub-polar areas such as western Norway, northern Swedish, British Columbia, and northern Canada[13,32,46,47]. Fluvial sediment in the Arctic induced by glacier recession and permafrost thaw has increased by many orders of magnitude[48–51]. If sediment fluxes increased as a result

of glacier recession, permafrost degradation, and extreme weather events, the consequences downstream would be serious and manifold, including increased carbon loading, reduced water quality and storage, bed aggradation, and rising flood stage. Thus, the ability of remote sensing technology to monitor and reconstruct fluvial sediment responses to progressive climate change has broader implications for human health, hydropower development, and ecosystem security.

## Methods

### Data and quality control

**In situ measurements.** To obtain reliable suspended sediment concentration (SSC) estimates for large rivers on the TP, we collected in situ water and sediment data from pristine headwater regions (Supplementary Table 1). These data were carefully selected to avoid anthropogenic impacts (e.g., dams, reservoirs, and hydropower stations)[1] and environmental noise[52], and to ensure they were consistent with satellite image acquisition frequencies. We utilized long-term, depth-integrated sampling approaches conducted by local hydrological stations, which are considered the most comprehensive method. Other sample data obtained from surface water, a single depth, and automatic pump samples were not considered due to the potential for sampling technique changes to introduce uncertainty in SSC calibrations[53]. In situ runoff and sediment data for the Chinese headwaters are sourced from the Hydrological Data Yearbook, the Ministry of Water Resources, China (http://www.mwr.gov.cn/sj/). Data for other international rivers come from the Water and Power Development Authority, Pakistan (http://www.wapda.gov.pk/) and refs. 34,41,42,54,55. The in situ observational data have undergone stringent quality control at the local monitoring stations. Further details can be found in the refs. 1,5.

**Satellite image collections.** The Landsat surface reflectance tier 1 image collections were obtained from the Landsat 5 Thematic Mapper (TM), Landsat 7 Enhanced Thematic Mapper Plus (ETM+), and Landsat 8 Operational Land imager (OLI) sensors[28]. The TM and ETM+ collections consist of four visible and near-infrared (VNIR) bands and two short-wave infrared (SWIR) bands that have been processed for orthorectified surface reflectance with a spatial resolution of 30 m/pixel. The OLI collections consist of five VNIR bands and two SWIR bands. To address the scan-line corrector failure in Landsat 7, we adopted the local linear histogram matching technique as recommended by the USGS, ensuring the integrity and completeness of Landsat imagery[56]. In light of variances in band configurations and responses among OLI, TM, and ETM+, we used designated band adjustment factors and algorithms to harmonize surface reflectance across them[57]. Using overlapping image dates and areas, we cross-calibrated Landsat 5, 7, and 8 reflectance values prior to preprocessing, ensuring consistency and eliminating systematic satellite biases.

The Landsat SR product provides essential information, including basic land cover classification, and quality assurance codes for each pixel in the 'pixel_qa' band. All image collections were atmospherically corrected and included a cloud, shadow, water, and snow mask, as well as a per-pixel saturation mask. A total of 145,908 Landsat images were applied to the Google Earth Engine (GEE, https://earthengine.google.com/) for further cloud removal, calibration, and mask extraction (Supplementary Fig. 1). Due to the limitations caused by Landsat's 16-day repeat period and challenges with mountainous shadows and clouds, we leveraged in situ observations from recent years to calibrate and interpolate the data for each region of interest (ROI) on the TP[58].

**Geographic and hydrography datasets.** We used the current best available global geographic and hydrological datasets to spatially couple and analyze sediment transport processes, including Multi-Error-Removed-Improved-Terrain (MERIT) DEM[59], MERIT Hydro[60], Global River Widths from Landsat (GRWL)[27], Global Land Ice Measurements from Space (GLIMS)[61], The GlObal geOreferenced Database of Dams (GOODD) dataset[62]. The MERIT DEM and Hydro, with a 3 resolution of arc-seconds, provided information on river flow direction, elevations, upstream/downstream links, and drainage area. We calibrated the extracted river masks and calculated the mean river width using the GRWL vector product, which has a width greater than 30 meters and was used to prevent seasonal or geomorphic disruption and evolution of each retrieved river. The GLIMS provides the most detailed global inventory of glaciers, we clipped it to obtain glacier area changes in the ROIs. We excluded the impacts of human activities by using GOODD, the largest georeferenced database of global dams, and by obtaining hydropower development data on the TP obtained from the International Hydropower Association (https://www.hydropower.org/status-report).

**Supplementary datasets.** Due to the limited in situ observations on the TP, we supplemented our analysis by collecting additional data to investigate the effects of climate change and geomorphic processes on the sediment yield and transport processes. These data included soil[63,64], land use[65], and vegetation changes[66], which are related to the sediment characteristics, as well as land surface temperature[67,68], precipitation[69], runoff[70,71], permafrost[72-74], glacier[35,75] and snow[76], which are related to thermal and hydraulic erosion. Note that the spatiotemporal resolution of these datasets was not entirely consistent with the sediment data retrieved from satellite imagery. Consequently, we endeavored to augment our dataset by incorporating on-site meteorological observations to the greatest extent possible. We also leveraged the fifth-generation European Center for Medium-Range Weather Forecasts (ECMWF) Reanalysis (ERA5) dataset as a supplementary source of meteorological intelligence in areas where on-site station data were not available.

### Preprocess of satellite imagery

Alpine environments are known for their complexity in optical images, with pixels containing various sources of environmental noise such as clouds, snow, ice, shadows, forests, and image artifacts[77]. Previous studies have attempted to extract surface water using Landsat images through techniques such as single-band thresholding and dual-band indexing, but these methods suffer from low accuracy and have limited applicability in alpine regions[52,78]. For example, in the western TP, these scenes exhibited diverse features due to elevation differences over short horizontal distances. To address these limitations, we employed multi-index methods to generate a reliable water mask for each image that overlapped one or more ROIs. Our approach included the use of several indices, such as the Normalized Difference Vegetation Index (NDVI), Normalized Difference Water Index (NDWI), Modified NDWI (MNDWI), and Automated Water Extraction Index (AWEI)[79], to improve accuracy and ensure adequate coverage of the targeted area.

To ensure high-quality water mask extraction from the satellite images, we began by eliminating images with excessive cloud cover (>20%). Next, we adopted the method outlined by ref. 80 to partition the TP region within the glacier-rich regions along the Himalayan mountain area (e.g., the Indus, Ganges, and upper Brahmaputra basins) and other cold regions. Regarding the main basins around the Himalayas, we segmented each ROI into areas above and below 665 m elevation and determined the optimal thresholds for each water index using the Otsu method[81] (Supplementary Fig. 11). Furthermore, for other cold regions, we employed the MNDWI and selected the Landsat band thresholds that distinguish between water and land based on ref. 53. To address the issue of sub-pixel level inundation changes or overestimating SSC during flood events, we utilized an automated method[82] to determine the water fraction in the extracted water masks and ensure the root mean square error (RMSE) of less than 7%.

To minimize the uncertainty caused by the varying geochemical properties and spectral signatures of water and sediment[53], we applied an unsupervised K-Means clustering approach to classify the optical properties of each river mask. We generated our ROIs by buffering the regions of each water mask by 90 m. Sub-pixel rivers (<30 m wide) were not preserved, although this may have caused spatial discontinuity. We focused only on the mainstreams and large tributaries of the large headwater basins on the TP, where water flow is continuous enough to obtain a long-term sediment sequence. For each image of the ROI, we calculated monthly spectral signatures using the subsets of all the image collections and generated the K-Means groupings by calling the Cluster function (ee.Clusterer.wekaKMeans{}) in GEE (Supplementary Fig. 12).

## Calculating suspended sediment concentration and flux

We compared the daily median pixel values of individual Landsat bands and band ratios. By setting a threshold for the minimum spectral feature distance from the K-Means cluster centroids, we assigned samples from each river's monthly subset to their respective K-Means groups. This dynamic assignment allowed for flexible, seasonally sensitive estimates of suspended sediment concentration (SSC), which strongly reduced errors and bias relative to using a single model[18]. Furthermore, we mapped the spatial distribution of the five optimum clusters, which can capture monthly observable river sections across the entire TP (Supplementary Fig. 12a). This approach additionally guarantees the classification of neighboring river sections into distinct categories based on variations in their spectral characteristics. Consequently, it ensures that the spatial variability of SSCs does not result from differences in the assessment of a single model. After normalizing values from distinct satellite sensor bands, marked variations in band values or their rations were evident among these clusters ($P < 0.001$, Supplementary Fig. 12b). We also plotted the true-color visuals of selected rivers based on their average RGB reflectance (Supplementary Fig. 14a). Subtle variations between each cluster will be showcased in terms of average true color juxtaposed with their respective SSC ranges.

Following Dethier et al.[18], we employed least-squares multiple regression to construct the SSC calibration model. After identifying the optimal five K-Means clusters, we used variables automatically selected from a list of spectral bands and band ratios using cross-validated Lasso Regression[83]. We aligned 75% of in situ SSC measurements with their corresponding band values or ratios, based on the consistent sampling coordinates and timings. These associations provided the foundation for further training the SSC calibration models, with a total of 32,656 individual matchups. For each of the five trained models, we employed an automated lambda penalty selection approach according to ref. [53], wherein we selected the model with the least number of explanatory variables that fell within one standard error of the most successful iteration of the model. This form of multiple regression eliminates low-power explanatory variables to avoid over-fitting and multi-collinearity. Furthermore, we reallocated river pixel clusters to location-matched satellite imagery for each model, enabling maximal extrapolation of local calibration models due to their consistent spectral characteristics in rivers[84]. This suite of algorithms aims to minimize the uncertainties inherent in satellite products, and the integration of machine learning further enhances the robustness of the models (Supplementary Fig. 11). The governing equation for the algorithm is:

$$\beta = \text{argmin}\left(\sum_{i=1}^{N}\left(SSC_i - \beta^T x_i\right)^2 + \lambda\left\|\beta\right\|1\right) \qquad (1)$$

where the subscript $i$ refers to the sequence number corresponding to each cluster, and SSC is the response variable. $x$ is the matrix of independent variables, representing the ratios or values between

K-Means grouping, for example, Band 4 and 5; band ratios B2/B1, B4/B3/B2. $\beta$ is the vector of regression coefficients, and $\lambda$ is the regularization parameter. The first term of the objective function is the squared error loss, and the second term is the L1 regularization term, which penalizes the sum of the absolute values of the regression coefficients. When $\lambda$ is larger, the model punishes the regression coefficients more heavily, and their values become smaller, or even zero. Therefore, Lasso regression can achieve feature selection and dimensionality reduction effects.

Using this algorithm, we deduced SSC values for specific moments by extracting surface reflectance band values and ratios from regions yet to be sampled. The resulting calibration models enabled us to generate maps of the median, mean, and summary of SSCs for each ROI. We amalgamated satellite-estimated SSCs with monthly average discharge data (in m³/s), predominantly derived from in situ station measurements. Yet, many rivers on the TP feature incomplete records, although the dataset we collected represents the most comprehensive long-term discharge dataset available for the region (Supplementary Table 1), as acknowledged in previous studies[1,12]. We also supplemented the discharge data by incorporating existing products specific to the entire TP[70,71], further enriched with ERA5 datasets. Temporally, we generated annual discharge time series ($Q_A$) based on monthly discharge records to align the temporal resolution with the satellite-estimated monthly SSC dataset. In cases of data gaps for river discharge, we filled the gaps using 3 or 5-year running annual averages.

In our analysis, we focused on the mainstreams and significant tributaries, characterized by a minimum river width of 90 meters, while smaller streams without available discharge data were excluded from consideration. To improve the precision of our annual sediment flux calculations, we determined the discharge-averaged SSC for each extractable river cross-section, aggregating these values into a one-year statistical average map (referred to as SSC_A, Eq. 2). Sediment flux was computed by multiplying the annual SSC (SSC_A) by the annual runoff ($Q_A$) for the river segments scaled to a 5 km river network. This calculation was performed for each river cross-section, following the downstream flow from the source to the outlet. This approach allowed us to present an extensive 36-year record for the major rivers across the TP. Many of these rivers lack contemporary published sediment observations, and some are even devoid of any historical records—remarkably, historical measurements align well with our reconstructed observations (Supplementary Fig. 16).

$$SSC_A = \frac{1}{Q_A}\sum_{i=1}^{12} SSC_i \times \bar{Q}_i \qquad (2)$$

where subscript $i$ refers to the monthly average value of the SSC and discharge, the bar denotes the whole-record average of that parameter, and subscript A refers to the annual average value of runoff.

## Validation and uncertainty

We aimed to minimize the uncertainty of our calibration models by utilizing unsupervised K-Means clustering, implementing river-specific corrections to the calibration models, and applying additional related parameters. We applied the procedures[53] to minimize both relative error and bias during their application (Eqs. 3–4; Supplementary Figs. 13 and 14). This error index offers clear interpretability[85], eschewing the pitfalls of asymmetry found in mean absolute error and the complexity of root mean square error on log-transformed values. Using the median, it resists distortions from outliers—a frequent occurrence in this extensive dataset owing to factors such as sampling errors, timing discrepancies between sampling and Landsat image

capture, image anomalies, and pixel misclassification.

$$rel.error = 10^{M(\lg|SSCestimated_i/SSCmeasured_i|)} - 1 \qquad (3)$$

$$model\ residual_i = \lg(SSCestimated_i) - \lg(SSCmeasured_i) \qquad (4)$$

where $SSC_{measured}$ and $SSC_{estimated}$ are taken from the holdout dataset that was not used for calibration, and where M denotes the median of SSC.

We validated models for each cluster using a total of 19,690 verified pairs of satellite-estimated SSCs and in situ measurements, achieving relative errors between 0.19 and 0.32 (Supplementary Fig. 13). We also calculated the median relative error for each cluster and the relative bias with respect to in situ data for each station (Supplementary Fig. 14). Site-specific validations between monitoring stations and satellite imagery further yielded a relative error of 0.24 and an average RMSE (root mean squared error) of 0.50 (Supplementary Fig. 15). We rigorously analyzed the model uncertainties, rooted in data quality, environment noise, long-term water-body changes, flood events, as well as the transferability of local calibration. We employed a suite of data quality controls and uncertainty assessment techniques to minimize uncertainties to acceptable thresholds. Further details can be found in the Supplementary Information.

### Sediment budget model
Tian et al.[86] developed a basin-wide Sediment Budget Method to systematically quantify sediment contributions from various tributaries and sections of the Yangtze River. We utilized this method's universal concept to quantify the sediment budget of each river we focused on, which can be monitored by Landsat satellites for almost four decades.

We improved the accuracy of point calculations (i.e., hydrological stations) by incorporating satellite-estimated sediment data, allowing us to capture the amount of deposition with greater precision. From the source to sink, we computed sediment flux variations for every 5 km segment along the river network, assessing whether sedimentation occurs or not based on Eq. 5. We centered on the mainstream and major tributaries, as signals from minor tributaries are susceptible to seasonal variations and shifts in erosion and deposition. Adhering to our flux computation guidelines, we cataloged the annual erosion and deposition patterns across the large TP rivers. We also quantified and mapped locations with either extant or potential for massive sediment deposition, combining these with the spatiotemporal SSC dataset we provided (Supplementary Fig. 8). In this method, proportional deposition ($P$) is a parameter used to simplify sediment deposition in river channels, which is defined by the following equations:

$$P = D/(SSF_a + SSF_b) \qquad (5)$$

where $SSF_a$ and $SSF_b$ are suspended sediment fluxes (Mt/yr) in river channels a and b, respectively. D represents the total amount of sediment deposition in the downstream river section after the convergence of channels a and b, and is calculated as the difference between the upstream and downstream sediment fluxes. We employed this coefficient to compute the annual erosion (or deposition) pattern in river systems, enabling us to assess the annual deposition events in the river channels. Stratifying the rivers into upper, middle, and lower sections based on their length, we systematically aggregated all measurable sediment fluxes for each respective year. This summation provided us with a comprehensive annual sediment erosion (or deposition) view of the entire river. Additional information on the Sediment Budget Method can be found in ref. 86.

### Analyzing trends in suspended sediment dynamics
We utilized the Mann–Kendall non-parametric trend test to investigate changes in annual SSC and SSF for each river, and we used a P-value to assess statistical significance. To facilitate statistical analysis, we established a sampling time interval from January 1, 1986, to December 31, 2021, as Landsat 5 began imaging the TP at the end of 1985 (https://developers.google.com/earth-engine/datasets/catalog/landsat). We chose an annual sampling interval due to the large-scale spatiotemporal resolution and the need for high precision and operating speed. We obtained single-day SSC data at ROIs. To account for glaciers and permafrost areas (e.g., Amu Darya, Indus basins), we restricted the retrieval time to the thaw periods (land surface temperature > 0°C). For areas with seasonally frozen ground or unfrozen areas, we did not impose time limitations. Therefore, we were able to reconstruct the sediment dynamics of each river for 36 years and analyze their trends.

## Data availability
Source data used in this analysis are publicly available. The data shown in the figures are available in the publications cited and at https://github.com/Waterloo255/TP-Riverine-Sediment. The suspended sediment concentration data generated in this study have been deposited in the Zenodo database (https://doi.org/10.5281/zenodo.10036364). Satellite images are available from the United States Geological Survey (https://earthexplorer.usgs.gov/) and Google Earth Engine (https://developers.google.com/earth-engine/datasets/catalog/landsat). Boundaries of glaciers and permafrost are available at the National Tibetan Plateau Data Center (https://data.tpdc.ac.cn/). The fifth-generation European Center for Medium-Range Weather Forecasts (ECMWF) Reanalysis (ERA5) dataset are available at https://www.ecmwf.int/en/forecasts/dataset/ecmwf-reanalysis-v5.

## Code availability
Code used to generate all analysis, figures, and tables can be found at https://github.com/Waterloo255/SSC-Retrieal-Algorithm.git. Other analytical codes generated in this paper are available upon request to the correspondence author.

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

## Acknowledgements

This study was supported by Joint Funds of the National Natural Science Foundation of China (U2240226; G.W.), the National Key Research and Development Program of China (No. 2022YFC3201702; S.S.), National Science Foundation of China (No. 42107062; C.S.), Joint Grant from Chinese Academy of Sciences–People's Government of Qinghai Province on Sanjiangyuan National Park (LHZX–2020–10; G.W.), and 2023 Sichuan University "From 0 to 1" Innovation Research Project (2023SCUH0096; C.S.). We highly appreciate Evan N. Dethier for his outstanding work in the field of global river sediment, which has significantly contributed to the advancement of this discipline.

## Author contributions

J.L. and G.W. conceived the study. J.L. wrote the original draft. C.S. and L.G. performed data processing and research analyses on the sediment changes, with the help of J.M. in the mapping and validation of remote sensing algorithms. J.L. and Y.W. analyzed and interpreted the driving forces of sediment yield and transport changes. G.W., C.S. and D.L. provided insightful feedback on methods and result interpretation. J.L., G.W., C.S., S.S., J.M., Y.W., L.G., and D.L. contributed to ideas and edits of subsequent revisions.

## Competing interests

The authors declare no competing interests.
