## [Peer Review File · Nature Communications]

Recent Intensified Erosion and Massive Sediment Deposition
in the Tibetan Plateau RiversREVIEWER COMMENTS

Reviewer #1 (Remarks to the Author):

The manuscript by Li et al. under review describes a regional-scale study in which sediment loads and fluxes were estimated for major rivers starting in the Tibetan Plateau. The study relied primarily on remote sensing imagery (Landsat) to investigate spatiotemporal trends in the headwaters of major Asian rivers, finding a general tendency towards increasing loads since the late 1980s, for which the authors (and others prior to them) make a strong argument that this is happening in association with global warming. Overall, the results of this study are of great importance to the understanding of some of the most important rivers on Earth, which support a disproportionately large fraction of the world's population.

This work is certainly of significance to the field, as it largely corroborates the message that Li et al (2021; well acknowledged in this manuscript) postulated in their estimates of sediment load increases using in situ data only. That said, the study does not present a completely new message, but instead, it presents an alternative -and more comprehensive- set of evidence for sediment load increases in the Tibetan Plateau based on remote sensing imagery.

The work presented supports the conclusions and claims stated. It was good to see that the focus of the study was on spatial and temporal trends, which is certainly the major gap for the Tibetan Plateau, rather than on drivers, which the authors skillfully controlled for by limiting the analysis to areas of little human development.

I could not find any evident flaws in the data analysis or interpretation of this study. However, I would encourage the authors to make their data and scripts open as soon as possible to allow reviewers and other scientists to reproduce and ratify their analysis.

The methodology of the study is sound, in general, although there are a few aspects that deserve some more clarity in the main text. With regards to the Landsat imagery, I was glad to read the SI subsection on Calibration and Uncertainty, but I wonder if some of these correct assertions about the uncertainty associated with remote sensing imagery analysis can be postulated in the discussion. Moreover, I did not see an explanation for how the authors dealt with the 2003 Landsat 7's Scan Line Corrector failure, and how this might have affected the trend analysis. Another important methodological aspect I was not clear about is the use of the calibration curves for the SSC calculation. The methods section broadly explains the process used, but it was not clear if all the curves shown in Fig S14 were used, or if only a selected group of them were selected based on some criteria. At least five of the curves have an error over 0.5.

An aspect of the work that I found particularly interesting and novel was the deposition and erosion patterns, which were not only quantified numerically, but also displayed geographically at a remarkably fine scale. These patterns, however, are purely based on a proportional deposition parameter (P), which is empirically derived from the SSF fluxes estimates. The evidence for such patterns would be much stronger if this empirical calculation was corroborated with a more physically-based representation of erosion and deposition, which are certainly well acknowledged in the geomorphology literature.

Mauricio E. Arias

Reviewer #2 (Remarks to the Author):

General comments

This paper estimated SSC and sediment flux across large rivers in the Tibetan Plateau using Landsat and then looked at trends in SSC/flux over time, over space from upstream to downstream, calculated a sediment budget to infer erosion/deposition, and compared flux estimates to factors such as glacial cover and land cover. Overall, this is a well-conceived study with excellent figures and interesting analyses.

I think this manuscript has great potential but requires more focus and refining. My two major recommendations are 1) using more precise and specific language. Often the language used was too general, or repeating the same concept with different words, that I could not understand the authors interpretations or framing. 2) Providing more details on methods when needed and providing quality checks to give us more confidence in the data and big interpretations made from these data. See more comments below.

Specific comments

Title: If the title is "rapid changes to erosion and sediment transport" I think it could be good to show that the rates of change in flux are faster than other studies from other parts of the world. OR consider changing the title to reflect the focus of the manuscript, which appeared to be more about spatial patterns in sediment flux and changes in sediment flux in cold regions than it was about rapid changes in erosion/transport.

Validation

The model validation process needs to be clearer. How many matchups were used to train models? Was 1 model built and applied to all points shown on the map? Was the same model used when applying to river pixels? Was there one model for each "cluster"? Is figure S14 the hold-out validation data that was used to evaluate the models? I find some of the relative error values strange considering the scatter shown in some of the plots (for example 48-Tuandukou. I wouldn't use a model with a validation that looked like this). Please elaborate on model validation and evaluation process and data and show the validation plots for each model used (if more than 1 model used).

If using more than 1 model, how can we be sure spatial variability in SSC or flux isn't due to differences in model predictions in adjacent river reaches/points/pixels?
How were differences in surface reflectance between landsat 5, 7, and 8 dealt with or tell us why this doesn't matter for your methods/study?

Fluxes

The flux calculation could benefit from more description. For example, what is the temporal resolution of the discharge data (daily?). The equation showed monthly SSC and Q averages were used which sounds good, but there will surely be months (particularly during monsoon) where there are no Landsat SSC observations. How was annual flux computed from sparse data and these biases in data quantity addressed?

The sediment budget model using the flux data could also use more description. Were the budget calculations made along a network of "river reaches"? IF so, what is the typically reach length used for the budget model? How can smaller tributaries without SSC or Q measurements be accounted for? What is the temporal scale of flux estimates used in the sediment budget model (long-term mean flux, annual flux, instantaneous flux from each image?)

-I was often confused when point based SSC/flux data, versus reach-based (if used at all?), versus pixel- based SSC was used in different analysis. Please be clear how data was aggregated over space and time for different analyses.

-In all analyses, how was temporal bias in Landsat observations dealt with (e.g. there will only be SSC

in non-rainy seasons for portion of TP with seasonal cloud cover, ice cover). Was data filtered to sites with representative data first? Was only summer season data used?

-The phrase "source to sink processes" is used many times and is a good phrase to frame the paper at the beginning, but I recommend getting more specific later on in the paper as this phrase is too general and can mean many things (e.g. are you talking about bed/bank erosion, deposition, flux, burial etc.)

Lines 25-27: Are the % increases in SSC and flux in comparison to "previously reported" studies or over the time period within this study? Please clarify. Also, if referring to previous studies, this is a difficult comparison to make in an abstract considering differences in methods/time periods etc. in previous studies. This result reported in the abstract didn't seem to be a focus and I couldn't find it in the manuscript. I would recommend reporting only major results from your study in abstract. What are the main one or two results you really want people to know?

Line 29-31: "Rapid shifts in the erosion-deposition patterns have regulated the sediment source to sink processes"... This sentence is an example of the type of wording that is frequently used but is too vague to have meaning. While I understand the general idea here, and its good to be general in the abstract, I recommend getting a little more specific as this sentence is difficult to interpret.

Lines 34-36: Consider revising this opening sentence and wording. 1) I assume "water energy" is referring to hydropower? Please clarify. 2) Hydropower is the only item in this list that is provided to people, while ecological function and sediment/carbon/nutrients are provided to ecosystems not people. Please revise this phrasing. 3) Is 2 billion people the population in the TP? If so please clarify and note TP rivers do not supply these things "worldwide" but only in the TP basin, so remove the word worldwide.

Line 41: What is a sedimentary effect?

Line 43: Is sediment availability refer to sediment supply? And availability in the landscape or within the river channel? Please be specific when possible.

Line 46-49: changes in sediment delivery in all environments is the balance of mobilization, transport, and deposition complicated by climate change. Consider getting more specific what this means in the context of "alpine cold environments".

Line 50: Note, sediment mobilization and transport are examples of "source-to-sink" processes, consider removing phrase source to sink here and/or be more specific.

Line 54: What is mean by "geomorphic feedback on these changes"? Consider being more specific or removing this phrase.

Line 85: This first paragraph seems to be more about spatial heterogeneity or do some of these statistics and sediment yields also refer to temporal trends? Please clarify and perhaps focus this paragraph on spatial patterns if that is your goal. So remove the word "spatiotemporal" if this is only spatial.

Line 137: "Furthermore, the thawing of permafrost has led to the expansion of thermokarst landscapes and hillslope mass wasting, which resulted in a relative higher SSCs where permafrost is developed on the TP". Is this an interpretation from the SSC data or relationships you found between thawing permafrost and SSC? Please clarify.

Figure 1. Are these data long-term average or annual averages as noted in caption?

Line 167-170: Consider rephrasing this sentence and be more specific. This is an example of a general statement found throughout the manuscript that is difficult to interpret because it's too general.

Line 222 and Table S2: Does the underestimation of sediment flux refer to comparison between yields in this study and other studies? If so, where are the sediment yields in Table 2 from other studies? Also if so, I assume other studies were not satellite based but hopefully they are based on surface SSC samples because satellite based SSC and flux will almost always be underestimated compared to a depth-integrated field based measurement. Just making sure your interpretation of underestimated fluxes is not caused by differences in data and satellite biases instead of the deposition processes mentioned.

Some papers that may be useful background or citations:

Balasubramanian S V et al 2020 Robust algorithm for estimating total suspended solids (TSS) in inland and nearshore coastal waters *Remote Sens. Environ.* 246 111768

Gardner et al 2023. Human activities change suspended sediment concentration along rivers. *Environ. Res. Lett.* 18 064032 <https://iopscience.iop.org/article/10.1088/1748-9326/acd8d8>

Claverie M, Vermote E F, Franch B and Masek J G 2015 Evaluation of the Landsat-5 TM and Landsat-7 ETM+ surface reflectance products *Remote Sens. Environ.* 169 390–403

Submitted to Nature Communications
Manuscript Number: NCOMMS-23-17241-T

Reviewers' Comments:

Reviewer #1:

Remarks to the Author:

The manuscript by Li et al. under review describes a regional-scale study in which sediment loads and fluxes were estimated for major rivers starting in the Tibetan Plateau. The study relied primarily on remote sensing imagery (Landsat) to investigate spatiotemporal trends in the headwaters of major Asian rivers, finding a general tendency towards increasing loads since the late 1980s, for which the authors (and others prior to them) make a strong argument that this is happening in association with global warming. Overall, the results of this study are of great importance to the understanding of some of the most important rivers on Earth, which support a disproportionately large fraction of the world's population.

This work is certainly of significance to the field, as it largely corroborates the message that Li et al (2021; well acknowledged in this manuscript) postulated in their estimates of sediment load increases using in situ data only. That said, the study does not present a completely new message, but instead, it presents an alternative -and more comprehensive- set of evidence for sediment load increases in the Tibetan Plateau based on remote sensing imagery.

The work presented supports the conclusions and claims stated. It was good to see that the focus of the study was on spatial and temporal trends, which is certainly the major gap for the Tibetan Plateau, rather than on drivers, which the authors skillfully controlled for by limiting the analysis to areas of little human development.

I could not find any evident flaws in the data analysis or interpretation of this study. However, I would encourage the authors to make their data and scripts open as soon as possible to allow reviewers and other scientists to reproduce and ratify their analysis.

The methodology of the study is sound, in general, although there are a few aspects that deserve some more clarity in the main text. With regards to the Landsat imagery, I was glad to read the SI subsection on Calibration and Uncertainty, but I wonder if some of these correct assertions about the uncertainty associated with remote sensing imagery analysis can be postulated in the discussion. Moreover, I did not see an explanation for how the authors dealt with the 2003 Landsat 7's Scan Line Corrector failure, and how this might have affected the trend analysis.

Another important methodological aspect I was not clear about is the use of the calibration curves for the SSC calculation. The methods section broadly explains the

process used, but it was not clear if all the curves shown in Fig S14 were used, or if only a selected group of them were selected based on some criteria. At least five of the curves have an error over 0.5.

An aspect of the work that I found particularly interesting and novel was the deposition and erosion patterns, which were not only quantified numerically, but also displayed geographically at a remarkably fine scale. These patterns, however, are purely based on a proportional deposition parameter (P), which is empirically derived from the SSF fluxes estimates. The evidence for such patterns would be much stronger if this empirical calculation was corroborated with a more physically-based representation of erosion and deposition, which are certainly well acknowledged in the geomorphology literature.

Mauricio E. Arias

Reviewer #2:

Remarks to the Author:

General comments:

This paper estimated SSC and sediment flux across large rivers in the Tibetan Plateau using Landsat and then looked at trends in SSC/flux over time, over space from upstream to downstream, calculated a sediment budget to infer erosion/deposition, and compared flux estimates to factors such as glacial cover and land cover. Overall, this is a well-conceived study with excellent figures and interesting analyses.

I think this manuscript has great potential but requires more focus and refining. My two major recommendations are 1) using more precise and specific language. Often the language used was too general, or repeating the same concept with different words, that I could not understand the authors interpretations or framing. 2) Providing more details on methods when needed and providing quality checks to give us more confidence in the data and big interpretations made from these data. See more comments below.

Specific comments:

Title: If the title is “rapid changes to erosion and sediment transport” I think it could be good to show that the rates of change in flux are faster than other studies from other parts of the world. OR consider changing the title to reflect the focus of the manuscript, which appeared to be more about spatial patterns in sediment flux and changes in sediment flux in cold regions than it was about rapid changes in erosion/transport.

Validation

The model validation process needs to be clearer. How many matchups were used to train models? Was 1 model built and applied to all points shown on the map? Was the

same model used when applying to river pixels? Was there one model for each “cluster” ? Is figure S14 the hold-out validation data that was used to evaluate the models? I find some of the relative error values strange considering the scatter shown in some of the plots (for example 48-Tuandukou. I wouldn't use a model with a validation that looked like this). Please elaborate on model validation and evaluation process and data and show the validation plots for each model used (if more than 1 model used).

If using more than 1 model, how can we be sure spatial variability in SSC or flux isn't due to differences in model predictions in adjacent river reaches/points/pixels?

How were differences in surface reflectance between landsat 5, 7, and 8 dealt with or tell us why this doesn't matter for your methods/study?

Fluxes

The flux calculation could benefit from more description. For example, what is the temporal resolution of the discharge data (daily?). The equation showed monthly SSC and Q averages were used which sounds good, but there will surely be months (particularly during monsoon) where there are no Landsat SSC observations. How was annual flux computed from sparse data and these biases in data quantity addressed? The sediment budget model using the flux data could also use more description. Were the budget calculations made along a network of “river reaches” ? IF so, what is the typically reach length used for the budget model? How can smaller tributaries without SSC or Q measurements be accounted for? What is the temporal scale of flux estimates used in the sediment budget model (long-term mean flux, annual flux, instantaneous flux from each image?)

-I was often confused when point based SSC/flux data, versus reach-based (if used at all?), versus pixel- based SSC was used in different analysis. Please be clear how data was aggregated over space and time for different analyses.

-In all analyses, how was temporal bias in Landsat observations dealt with (e.g. there will only be SSC in non-rainy seasons for portion of TP with seasonal cloud cover, ice cover). Was data filtered to sites with representative data first? Was only summer season data used?

-The phrase “source to sink processes” is used many times and is a good phrase to frame the paper at the beginning, but I recommend getting more specific later on in the paper as this phrase is too general and can mean many things (e.g. are you talking about bed/bank erosion, deposition, flux, burial etc.)

Lines 25-27: Are the % increases in SSC and flux in comparison to “previously reported” studies or over the time period within this study? Please clarify. Also, if referring to previous studies, this is a difficult comparison to make in an abstract considering differences in methods/time periods etc. in previous studies. This result reported in the abstract didn't seem to be a focus and I couldn't find it in the manuscript. I

would recommend reporting only major results from your study in abstract. What are the main one or two results you really want people to know?

Line 29-31: “Rapid shifts in the erosion-deposition patterns have regulated the sediment source to sink processes” ... This sentence is an example of the type of wording that is frequently used but is too vague to have meaning. While I understand the general idea here, and its good to be general in the abstract, I recommend getting a little more specific as this sentence is difficult to interpret.

Lines 34-36: Consider revising this opening sentence and wording. 1) I assume “water energy” is referring to hydropower? Please clarify. 2) Hydropower is the only item in this list that is provided to people, while ecological function and sediment/carbon/nutrients are provided to ecosystems not people. Please revise this phrasing. 3) Is 2 billion people the population in the TP? If so please clarify and note TP rivers do not supply these things “worldwide” but only in the TP basin, so remove the word worldwide.

Line 41: What is a sedimentary effect?

Line 43: Is sediment availability refer to sediment supply? And availability in the landscape or within the river channel? Please be specific when possible.

Line 46-49: changes in sediment delivery in all environments is the balance of mobilization, transport, and deposition complicated by climate change. Consider getting more specific what this means in the context of “alpine cold environments” .

Line 50: Note, sediment mobilization and transport are examples of “source-to-sink” processes, consider removing phrase source to sink here and/or be more specific.

Line 54: What is mean by “geomorphic feedback on these changes” ? Consider being more specific or removing this phrase.

Line 85: This first paragraph seems to be more about spatial heterogeneity or do some of these statistics and sediment yields also refer to temporal trends? Please clarify and perhaps focus this paragraph on spatial patterns if that is your goal. So remove the word “spatiotemporal” if this is only spatial.

Line 137: “Furthermore, the thawing of permafrost has led to the expansion of thermokarst landscapes and hillslope mass wasting, which resulted in a relative higher SSCs where permafrost is developed on the TP” . Is this an interpretation from the SSC data or relationships you found between thawing permafrost and SSC? Please clarify.

Figure 1. Are these data long-term average or annual averages as noted in caption?

Line 167-170: Consider rephrasing this sentence and be more specific. This is an example of a general statement found throughout the manuscript that is difficult to interpret because it's too general.

Line 222 and Table S2: Does the underestimation of sediment flux refer to comparison between yields in this study and other studies? If so, where are the sediment yields in Table 2 from other studies? Also if so, I assume other studies were not satellite based but hopefully they are based on surface SSC samples because satellite based SSC and flux will almost always be underestimated compared to a depth-integrated field based measurement. Just making sure your interpretation of underestimated fluxes is not caused by differences in data and satellite biases instead of the deposition processes mentioned.

Some papers that may be useful background or citations:

Balasubramanian S V et al 2020 Robust algorithm for estimating total suspended solids (TSS) in inland and nearshore coastal waters *Remote Sens. Environ.* 246 111768

Gardner et al 2023. Human activities change suspended sediment concentration along rivers. *Environ. Res. Lett.* 18 064032 <https://iopscience.iop.org/article/10.1088/1748-9326/acd8d8>

Claverie M, Vermote E F, Franch B and Masek J G 2015 Evaluation of the Landsat-5 TM and Landsat-7 ETM+ surface reflectance products *Remote Sens. Environ.* 169 390 – 403

We have thoroughly addressed the referee's comments in the revised manuscript. A detailed, point-by-point response to each comment can be found within this document.

Reviewers' Comments

Reviewer #1:

Remarks to the Author:

The manuscript by Li et al. under review describes a regional-scale study in which sediment loads and fluxes were estimated for major rivers starting in the Tibetan Plateau. The study relied primarily on remote sensing imagery (Landsat) to investigate spatiotemporal trends in the headwaters of major Asian rivers, finding a general tendency towards increasing loads since the late 1980s, for which the authors (and others prior to them) make a strong argument that this is happening in association with global warming. Overall, the results of this study are of great importance to the understanding of some of the most important rivers on Earth, which support a disproportionately large fraction of the world's population.

Response: We are grateful for your time and effort in reviewing our manuscript and for recognizing the significance of our study on the sediment dynamics of major rivers originating from the Tibetan Plateau (TP). We will continue to refine our analyses and presentation based on the constructive feedback received during this review process. Your insights and comments are invaluable in ensuring that our work meets the highest standards of scientific rigor and clarity.

This work is certainly of significance to the field, as it largely corroborates the message that Li et al (2021; well acknowledged in this manuscript) postulated in their estimates of sediment load increases using in situ data only. That said, the study does not present a completely new message, but instead, it presents an alternative -and more comprehensive- set of evidence for sediment load increases in the Tibetan Plateau based on remote sensing imagery.

Response: We are glad that our study resonates with the findings of Li et al. (2021). Our intention was indeed to build upon their foundational work and offer a broader perspective using remote sensing techniques. By cross-referencing in situ data with satellite imagery, we aim to present a more holistic understanding of sediment dynamics on the TP. Furthermore, we hope this multi-faceted approach enriches the existing body of knowledge and offers a robust foundation for future research endeavors.

The work presented supports the conclusions and claims stated. It was good to see that the focus of the study was on spatial and temporal trends, which is certainly the major gap for the Tibetan Plateau, rather than on drivers, which the authors skillfully

controlled for by limiting the analysis to areas of little human development.

Response: Yes! It's encouraging to see your acknowledgment of our emphasis on spatial and temporal trends. One of the most fundamental constraints in sediment research in the TP and analogous cold regions is the paucity of data. Gathering in situ samples is not only expensive and infrequent but is also often shielded from public access (Dethier et al., 2020; East and Sankey, 2020; Zhang et al., 2022). Such limitations hamper the exploration of change trends on fluvial sediment across broader spatiotemporal scales. Consequently, many prevailing studies, anchored primarily in field data, are limited in their scope, often constrained to minor or intermediate basin scales (Ali and De Boer, 2007; Li et al., 2023; Shi et al., 2022).

Driven by these challenges, our core aim has been to leverage the capabilities of remote sensing and big data analytics to meticulously map the sediment dynamics across the Tibetan Plateau's riverine networks. Furthermore, minimizing the influence of human activities using technological interventions is paramount. The construction of dams and anthropogenic disruptions in land-use patterns significantly curtail sediment transport—a phenomenon extensively documented in the downstream regions of major global river basins (Syvitski et al., 2022; Tian et al., 2021; Wang et al., 2016).

I could not find any evident flaws in the data analysis or interpretation of this study. However, I would encourage the authors to make their data and scripts open as soon as possible to allow reviewers and other scientists to reproduce and ratify their analysis.

Response: We wholeheartedly agree with the importance of transparency and reproducibility in scientific research. We are committed to the open science ethos and understand its value in fostering collaboration, trust, and innovation within the scientific community.

In line with your request, we are delighted to introduce our comprehensive suspended sediment concentration (SSC) dataset, capturing the fluvial dynamics of the Tibetan Plateau over the span of 1986-2021. This annual-scale dataset, archived in the GeoTIFF format with > 513 GB, spans the intricate river systems meticulously delineated using our bespoke extraction strategy (detailed in the Method section).

Alongside this dataset, we have provided essential processing scripts, which encompassed: (a) A tailored application of the K-Means clustering algorithm for the TP's riverine systems (Java Script); (b) The SSC Retrieval Algorithm (Java Script and R); (c) Advanced algorithms for evaluating fluvial sediment flux and related parameters analyze (Python).

We agree with your insights and are confident that this contribution will not only streamline the review process but also provide a valuable resource for the wider scientific community to leverage.

We have made the data and scripts open at the following website:

<https://doi.org/10.5281/zenodo.10036364>

<https://github.com/Waterloo255/SSC-extract-and-calibration.git>

The methodology of the study is sound, in general, although there are a few aspects that deserve some more clarity in the main text. With regards to the Landsat imagery, I was glad to read the SI subsection on Calibration and Uncertainty, but I wonder if some of these correct assertions about the uncertainty associated with remote sensing imagery analysis can be postulated in the discussion. Moreover, I did not see an explanation for how the authors dealt with the 2003 Landsat 7's Scan Line Corrector failure, and how this might have affected the trend analysis.

Response: To address the reviewer's comment. We take efforts on the following points :

1. Landsat Imagery Uncertainty in the Discussion: We acknowledge the significance of directly confronting the uncertainties tied to remote sensing imagery within the core content of our manuscript. While we have detailed the Calibration and Uncertainty in the Supplementary Information, we appreciate your recommendation to embed pivotal statements regarding this uncertainty into the discussion. Consequently, we've appended a subsection at the Method sections, titled 'Calibration and Uncertainty'. This inclusion aims to offer readers a more immediate insight into both the constraints and strengths of our methodology, thereby enriching the depth of our discourse.

2. Addressing the Landsat 7's Scan Line Corrector (SLC) Failure: We appreciate your astute observation on this matter. It was an oversight on our part to not highlight this issue in our preliminary draft. We employed the 'local linear histogram matching technique'—a neighborhood pixel regression method—provided officially by the USGS (Scaramuzza et al., 2004), to address and rectify the SLC gaps in Landsat 7 imagery. To elucidate our methodology further, we have crafted a dedicated algorithm to demonstrate how we addressed the Landsat 7 SLC issue within the Google Earth Engine (GEE) framework (See lines 367-374).

You can access and execute our script via the following link. Please note that prior registration with GEE is necessary to run this code on the cloud platform. Additionally, we have refined our manuscript on Landsat preprocessing (See *Data and quality control*) to succinctly convey our solution to this challenge to our readers.

https://code.earthengine.google.com/?scriptPath=users%2FKingDr%2FSSC_TP%3AL7_SCL

Another important methodological aspect I was not clear about is the use of the calibration curves for the SSC calculation. The methods section broadly explains the

process used, but it was not clear if all the curves shown in Fig S14 were used, or if only a selected group of them were selected based on some criteria. At least five of the curves have an error over 0.5.

Response: We appreciate the reviewer's astute observation. While the reviewer noted that we provided a broad explanation of our computational process, we recognize the importance of clarity and precision in our description. We used K-Means clustering to categorize the river pixels into five classes and trained machine learning algorithms on the derived surface emissivity values and in situ SSCs available (See Methods, *Calculating suspended sediment concentration and flux*). Afterward, we collectively employed cluster validation (Fig. S12-13), uncertainty analysis (Fig. S14), and site-specific validation (Fig. S15-16) to individually validate our results. In the revision, we have delineated these processes with greater clarity in the Methods section (refer to the revised section *River cluster grouping, Calculating suspended sediment concentration and flux, and Calibration and uncertainty*). We have introduced several paragraphs and a new figure (supplementary Fig. S13) to how we deal with uncertainty and elucidate the robustness of our validation process.

In Fig S14 (Fig. S15 in the upload version), we present 28 validation curves, with each curve corresponding to *in situ* data from a specific sampling location (Consistent with the spatial coordinates and sampling periods during satellite acquisitions). These curves are intended to validate the results of each SSC calibration model. For further details, please refer to Fig. S15 in the revised version, titled 'Validation results between satellite-estimate SSC and in situ SSC sourced from measurement records of 28 hydrological stations on the TP.' The creation of these curves serves the purpose of confirming the robustness of our model in reproducing *in situ* data. Given the disparity in the time span of the *in situ* data from 1986 to 2021, we carried out validations for three distinct sampling periods corresponding to satellite acquisitions to ensure data consistency.

In relation to five curves have an error over 0.5, it is important to note that our chosen error calculation (Equation. 3 in the Main Text) is based on the relative error formula employed by Morley et al. (2018). This approach offers the benefit of straightforward interpretation (in contrast to the root-mean-square error on log-transformed values) and sidesteps the asymmetry issues associated with using mean absolute error.

Furthermore, we meticulously scrutinized the raw data in Fig. S14 (Fig. S15 in the upload version), excluding samples not determined by the depth-integration method. We rigorously re-plotted the image on a 1:1 scale (in earlier versions, the x and y axes were not perfectly aligned). Nonetheless, we have endeavored to minimize this range as much as possible, primarily due to difference in basin sizes and Landsat imagery, with the value distribution that currently spans from 0.11 to 0.61. We have also incorporated the widely adopted root-mean-square error (RMSE) to provide readers with a clearer representation of our validation results for in situ station data.

Additionally, compared to previous studies, for instance, in a comprehensive survey of 414 large river estuaries globally conducted by Dethier et al. (2022), the relative error values ranged between 0.05 and 6.70 (you can refer the Excel spreadsheet available at the following link <https://github.com/evandethier/satellite-ssc/tree/master/landsat-calibration/landsat-57-standalone-calibrations>). We believe that our calibration results fall within an acceptable error range. For instance, concerning the calibration at a few relatively high-SSC sites, their *in situ* measured SSC is significantly higher compared to sites with relatively smaller relative errors.

An aspect of the work that I found particularly interesting and novel was the deposition and erosion patterns, which were not only quantified numerically, but also displayed geographically at a remarkably fine scale. These patterns, however, are purely based on a proportional deposition parameter (P), which is empirically derived from the SSF fluxes estimates. The evidence for such patterns would be much stronger if this empirical calculation was corroborated with a more physically-based representation of erosion and deposition, which are certainly well acknowledged in the geomorphology literature.

Response: We deeply appreciate your recognition of our effort in presenting the erosion and deposition patterns in both a quantitative and geographically detailed manner. For the first time, we have quantified erosion and deposition patterns in the TP rivers using the sediment budget model. The current frontier in this domain of water and sediment science has been limited to acknowledging the fact that sediment discharge increases at specific sites, and as you mentioned in the above comment, to using mathematical and statistical methods to analyze how climate change might be contributing to the increase in sediment discharge (Li et al., 2021a; J. Li et al., 2023; Shi et al., 2022).

Your insight regarding the proportional deposition parameter (P) being solely empirically derived from the SSF estimates is astute. While physics-based assessments of riverine erosion or deposition offer a more comprehensive view of the TP's sedimentary landscape, however, undertaking such an endeavor is undoubtedly substantial and extends well beyond the scope of this study. Even the most common WEPP model, developed by the United States Department of Agriculture (USDA), is currently the most widely used numerical simulation software for soil erosion. The WEPP model employs a series of modules to describe the soil erosion process, with the most crucial module being the Universal Soil Loss Equation (USLE) equation. However, it is well known that both the USLE and RUSLE are empirically derived, and researchers often modify empirical parameters based on different study locations (Islam, 2022; Teng et al., 2018).

Furthermore, existing sediment-yield models are mostly empirical or conceptual erosion modules (for example, soil erosion modules in SWAT, WBMsed, HydroTrend, BQART and SAT). Physics-based models are rare (for example, Water Erosion

Prediction Project) and only marginally account for the temperature-dependent erosional processes in cryospheric basins (Zhang et al., 2022).

We take the widely-used physis-based' stream power law and Navier-Stokes model as examples (Harel et al., 2016), expressed as

$$E = KA^mS^n \quad (1)$$

Where E is the erosion rate (mm/yr); K is an erodibility coefficient that encompasses the influence of climate, lithology, and sediment transport processes; A is the upstream drainage area; and $S = -\partial z/\partial x$ is the local channel slope with z the elevation. The slope and drainage area exponents, respectively n and m, are empirical constants.

The revised Navier-Stokes model is expressed as

$$\partial u/\partial t + (u \cdot \nabla)u = -1/\rho \cdot \nabla p + \nu \nabla^2 u + F_s \quad (2)$$

Where u is the fluid velocity vector; t is time; p is pressure; and ρ is fluid density. ν is the dynamics viscosity (kinematic viscosity). F_s represents the contribution of sediment particles to the fluid concentration.

Attempting to utilize these formulas to discern whether river channels on the TP are undergoing erosion or sediment deposition is a substantial undertaking. Even without considering the adaptations of the equation from previous studies regarding factors such as channel width, sediment supply, fluid force, and sediment threshold adjustments, several challenges arise:

- 1) Determining the standard erosivity coefficients for each headwater basin on the TP. This determination would necessitate the incorporation of localized parameters such as rock type, climate, and vegetation, based on field observations and existing records.
- 2) Identifying the empirical exponents m and n for different basin. Or identifying the regional fluid drag, bed friction, and mutual collisions.
- 3) For permafrost or seasonally frozen areas, it would be essential to modify the equation to account for climatic variables such as temperature and precipitation.

This work is undoubtedly immense, and any modification to the river erosion equation remains a matter of debate within the academic community regarding its comprehensive applicability to the TP region (Zhang et al., 2022, 2021). In the early stages of our research, we contemplated how to determine river channel erosion and deposition, and certainly considered the use of more established and robust equations to underpin our core arguments. Fortunately, as we began to explore this avenue, Tian and colleagues, following peer review, published their sediment balance algorithm for determining erosion/deposition in the Yangtze River in Water Resources Research (Tian et al., 2021). This offered us a glimpse into the feasibility of calculating river sediment status across a vast region in a manner that is not only succinct and computationally efficient but also stable and robust.

Of course, we resonate deeply with the issues you've highlighted. These indeed represent significant avenues for advancing the frontier of sediment transport mechanisms in alpine regions. Based on physically based sediment transport equations, this is also a focal area and frontier of our future research. One of the reasons we have made our dataset publicly available is with the hope that researchers at large can access a substantial volume of data, thereby facilitating the development of more robust models for cold-region sediment transport processes.

In conclusion, we believe that, for large-scale sediment transport in cold regions, the use of the Sediment Budget Model (SBM) is currently the most robust and computationally straightforward method. We have not attempted to employ alternative methods to reevaluate this issue. However, based on your concern, we have incorporated the limitations of our study, the uncertainties associated with satellite methods, and the outlook into our revised manuscript (See Supplementary Information, *Limitations and Perspectives*). We are immensely grateful for your invaluable guidance.

Reviewer #2:

Remarks to the Author:

General comments:

This paper estimated SSC and sediment flux across large rivers in the Tibetan Plateau using Landsat and then looked at trends in SSC/flux over time, over space from upstream to downstream, calculated a sediment budget to infer erosion/deposition, and compared flux estimates to factors such as glacial cover and land cover. Overall, this is a well-conceived study with excellent figures and interesting analyses.

I think this manuscript has great potential but requires more focus and refining. My two major recommendations are 1) using more precise and specific language. Often the language used was too general, or repeating the same concept with different words, that I could not understand the authors interpretations or framing. 2) Providing more details on methods when needed and providing quality checks to give us more confidence in the data and big interpretations made from these data. See more comments below.

Response: We extend our gratitude to you for astutely summarizing and recognizing the essence of our research findings. We have meticulously addressed each of your observations and recommendations. As a result, we feel that our manuscript's clarity, context, and robustness have been substantially bolstered, particularly in the main text. Additionally, the Supplementary Information in our resubmission provides added depth and detail.

Considering your primary concerns and specific comments, we subjected the manuscript to a rigorous review. While the revisions span the entirety of the main text, we prioritized enhancements in the Introduction due to concerns related to linguistic clarity. Furthermore, we have augmented the Methods section with supplemental information. Notably, we introduced a comprehensive paragraph detailing the validation of the SSC algorithm (See lines 530-555) and furnished a synopsis of the limited in-situ observations, which underscores the good alignment of our data with ground-based measurements.

Specific comments:

Title: If the title is “rapid changes to erosion and sediment transport” I think it could be good to show that the rates of change in flux are faster than other studies from other parts of the world. OR consider changing the title to reflect the focus of the manuscript, which appeared to be more about spatial patterns in sediment flux and changes in sediment flux in cold regions than it was about rapid changes in erosion/transport.

Response: Reviewer 2 has pointed out that our work primarily describes the spatiotemporal trends of sediment concentration and flux, as well as calculations regarding sediment equilibrium in terms of erosion and deposition. This is accurate;

indeed, our main focus was on utilizing satellite big data techniques to reconstruct the sediment dynamics of the entire Tibetan Plateau over a 36-year period, and rigorously identifying the challenges presented by erosion-deposition transitions within the river systems of the Plateau. However, we do not wish to mislead or exaggerate our conclusions, and thus, we have revised the title to be “Recent Intensified Erosion and Massive Sediment Deposition in Tibetan Plateau Rivers”. We believe the updated title more accurately and clearly reflects the main contributions of our paper.

Validation

The model validation process needs to be clearer. How many matchups were used to train models? Was 1 model built and applied to all points shown on the map? Was the same model used when applying to river pixels? Was there one model for each “cluster” ?

Response: We concur with the reviewer's observation regarding the calibration issues in our initial manuscript. We first employed K-Means clustering to categorize the river pixels (with a buffer > 90m) into five classes. Subsequently, we combine the surface reflectance band values or band ratios obtained from these five clusters with corresponding in-situ SSC measurements, following Equation 1, to train five calibration models that depend on their spectral characteristics. In the new version of the manuscript (lines 473-480), we explicitly state that we used a total of 32,656 pairs to train our five calibration models. Subsequently, we employ an automated lambda penalty selection approach (Dethier et al., 2020) to iteratively compute and converge the models (lines 477-480). Then, we extrapolate these five models to river pixels of the respective K-Means classifications, achieving an extension of SSC calibration models (lines 482-484).

We recognize that our previous description was not comprehensive enough to allow other researchers or readers to replicate our algorithm. Therefore, we have amended our representation in the methods section and have detailed the exact algorithmic procedure in the manuscript (See Methods, *River cluster grouping* and *Calculating suspended sediment concentration and flux*). In addition to this adjustment, we've also incorporated comparison charts of the validation results from the five calibrated models against the *in situ* SSC measurements (with a total of 19690 corresponding pairs, Supplementary Fig. S13).

Is figure S14 the hold-out validation data that was used to evaluate the models? I find some of the relative error values strange considering the scatter shown in some of the plots (for example 48-Tuandukou. I wouldn't use a model with a validation that looked like this). Please elaborate on model validation and evaluation process and data and show the validation plots for each model used (if more than 1 model used).

Response: Fig. S14 represents 28 validation plots of the outputs against the *in situ* data from the monitoring stations. In this figure, the sampling times and locations are

meticulously aligned with the data extracted from satellite imagery. In response to the previous comment, we have provided a concise summary of the development and training of our SSC calibration models. Additionally, we used cluster validation (Fig. S12-13, available in the new version), uncertainty analysis (Fig. S14), and site-specific validation (Fig. S15-16) to individually verify our outcomes. A new subsection, "Validation and Uncertainty" has been included in the Methods section. Within this subsection, we address two key aspects: 1) how we control model uncertainty, and 2) a more comprehensive description of our validation and evaluation processes (See lines 530-555).

Regarding Fig. S14 (now Fig. S15 in the updated version), we acknowledge that in the original figure, the X and Y-axes were not aligned correctly, resulting in discrepancies. For instance, in the original supplementary document's Fig. S14 for the Tuandukou site, the y-axis began at 0.6, while the x-axis started at 0.7. We greatly appreciate your keen observation in identifying this oversight. It's essential to note that our initial purpose in presenting this figure was to demonstrate the disparity between satellite-estimated SSC and in situ measurements to our readers. To provide readers with a more precise and intuitive representation, we meticulously reevaluated the original data and reconstructed the figure (see Fig. S15).

Additionally, in line with your suggestion, we have incorporated validation plots for the five models we developed (refer to Figure S13). We anticipate that this enhancement will provide readers with a more precise and intuitive representation of the robustness of our findings.

If using more than 1 model, how can we be sure spatial variability in SSC or flux isn't due to differences in model predictions in adjacent river reaches/points/pixels?

Response: We appreciate your pointing out this concern. The very reason we employed K-Means optical clustering is to address this issue. In previous studies that used satellite remote sensing to invert SSC, the common approach was to train models by comparing observed SSC (a single sampling point) with corresponding band pixel values from imagery (Balasubramanian et al., 2020; Overeem et al., 2017). These models are empirical and, if applied to extrapolate across the entire basin, can easily result in the inaccuracies you mentioned among neighboring river reaches/points/pixels. If we attempted to train our provided sediment data from 28 stations using a single model, the error would significantly surpass the error associated with developing calibration models based on K-Means clustering. To provide a point of reference, Dethier et al. (2020) introduced this method, demonstrating that the relative error using the clustering approach is 0.49 while employing a single model with the base method results in a substantially higher relative error of 0.97.

By utilizing optical clustering of river pixels (which you can clearly see in Supplementary Fig. S12 for spatial distribution and in Fig. S14a for true color display)

combined with machine learning, we have avoided interference from adjacent river pixels. In lines 454-499, We have expounded upon two pivotal facets: 1) our assignment of river pixels to site-specific river locations through K-Means clustering, and 2) the spatial visualization of these assignments (for a detailed spatial distribution of each cluster, please refer to S12).

The calibration models for each river are fashioned based on the dynamic allocations generated by K-Means. Moreover, models trained for diverse K-Means classifications associated with distinct rivers adhere to identical fundamental principles, as delineated in Equation (1) found on line 488. This approach ensures that even neighboring river segments are grouped into different categories, reflecting disparities in the spectral characteristics of river pixels, including band values and band ratios. Hence, this approach ensures that the spatial variability of SSC is primarily governed by its intrinsic optical characteristics, rather than variations in predictions or parameters of different models. This is because all five models operate based on the same underlying principle.

How were differences in surface reflectance between landsat 5, 7, and 8 dealt with or tell us why this doesn't matter for your methods/study?

Response: We thank the reviewer for this concern. Indeed, Landsat 8's Operational Land Imager (OLI) has a slightly different band configuration and response compared to the Thematic Mapper (TM) on Landsat 5 and the Enhanced Thematic Mapper Plus (ETM+) on Landsat 7. To account for this, we used established band adjustment factors and algorithms to make the reflectance values from OLI consistent with those from TM and ETM+. Furthermore, before conducting any analysis, we cross-calibrated the surface reflectance values from Landsat 5, 7, and 8 using overlapping image dates and regions. This helped in adjusting for any systematic biases among the satellites and brought the reflectance values to a common scale.

We acknowledge the importance of providing a thorough and comprehensive description in the Methods section. We have succinctly added the necessary details to the Main Text (*See Data and quality control, 2. Satellite image collections*, lines 369-372). We have also added the references you specified that are relevant to this recommendation (Claverie et al., 2015). We have also supplemented with the following code to facilitate your review. The following link is provided to demonstrate how this method controls the differences in surface reflectance of Landsat 5, 7, and 8 satellites within GEE, particularly addressing the disparities between Landsat 8 bands and those of Landsat 5 and 7.

<https://code.earthengine.google.com/3ba4b4523b0b107c609e64e6ad2cf6c7?noload=true>

Fluxes

The flux calculation could benefit from more description. For example, what is the

temporal resolution of the discharge data (daily?). The equation showed monthly SSC and Q averages were used which sounds good, but there will surely be months (particularly during monsoon) where there are no Landsat SSC observations. How was annual flux computed from sparse data and these biases in data quantity addressed?

Response: To address the reviewer's concern, we have included additional information in the methods section of the revised document. Specifically, we have included an extensive description discussing the runoff product we employed (X. Li et al., 2023; Wang et al., 2021) and how it is coupled with our inferred SSC product (see lines 499-528). Detailed explanations regarding the temporal and spatial resolution of our flux calculations have been provided. We also emphasized our approach to handling sparse data and compared our flux product with *in situ* measurements from multiple sites (Fig. S16). We believe these elaborations are comprehensive, allowing readers to clearly replicate our methodology (See Methods, *Calculating suspended sediment concentration and flux*).

In the vast expanse of the TP, spanning approximately 2.5 million square kilometers (a mere 185 x 180 square kilometers covered by a single Landsat image), the Landsat records have preserved a substantial archive of cloud-free observations, even through the monsoon season. Further details are available in our supplementary Fig. S18, which documents the monthly image acquisitions and variations in monthly satellite-estimated SSC spanning the period from 1986 to 2021. Note that the frequency of cloud-free Landsat images is reduced during the monsoon season compared to non-monsoon months, however, this discrepancy remains relatively minor (Fig. S18a).

Moreover, note that satellite-estimated SSC and sediment flux dynamics are significantly higher in monsoon (summer) than in non-monsoon (winter, Fig. S18b). This is because erosion and related surface processes in cold region are stationary (J. Li et al., 2023; Zhang et al., 2022). In the first step of our algorithms, 'Preprocess of satellite imagery', we diligently eliminate elements such as frozen rivers, ice, and snow. As a pivotal adjustment, we have introduced the concept of the runoff weighting coefficient (as defined in Equation 2) in the processing of SSC samples. This coefficient effectively filters out the aforementioned mismatched data, affording us a holistic perspective of the annual mean distribution, referred to as the SSC_A . This method was proposed by Dethier et al, which has been applied to the SSC assessment of global estuaries and tropical rivers (Dethier et al., 2023, 2022). In those regions where monsoon seasons bring heavy rainfall and significant cloud interference (especially in the case of Amazon River), Dethier et al., (2023) successfully managed to capture the annual sediment dynamics in their study regions. Therefore, we believe that the SSC and flux data we obtained are complete and can characterize the data level of sediment discharge throughout the year.

The sediment budget model using the flux data could also use more description. Were the budget calculations made along a network of "river reaches" ? IF so, what is the

typically reach length used for the budget model? How can smaller tributaries without SSC or Q measurements be accounted for? What is the temporal scale of flux estimates used in the sediment budget model (long-term mean flux, annual flux, instantaneous flux from each image?)

Response: To address these concerns, we have elucidated the process of our flux data calculations. Using GEE, we refined the spatial resolution of the sediment transport model. The original Sediment Budget Model (SBM) relies on sediment station data, which are often distributed over tens or even hundreds of kilometers. We calculated and ran the model for each segment along a 5km river network from source to basin outlet. As we have consistently emphasized in the manuscript, our focus is on major rivers and their tributaries. Streams that are smaller than three pixels of the satellite's resolution (90m, using a buffer setting for most remote sensing inversions of rivers with Landsat products) – the smaller tributaries you referred to – were not considered. This is because they might be in mixed-pixel areas where land and water intersect, or the river width might not be sufficient for satellite capture. We have also included statements detailing the temporal and spatial resolution of this model (See Methods, *Calculating suspended sediment concentration and flux*, lines 557-586).

-I was often confused when point based SSC/flux data, versus reach-based (if used at all?), versus pixel- based SSC was used in different analysis. Please be clear how data was aggregated over space and time for different analyses.

Response: As mentioned in the revised methods section above, we have elaborated on our utilization of various SSC datasets. Note that for point-scale SSC/flux data, especially those based on *in situ* measurements, we employed them to train and calibrate our model (Supplementary Fig. S13-16). Once the model passed validation, we utilized satellite-derived pixel-based data to compute averages for 5km river segments. These averages were then incorporated into subsequent sediment balance model calculations. We have also taken into account the temporal resolution (See Methods, *Calculating suspended sediment concentration and flux*).

-In all analyses, how was temporal bias in Landsat observations dealt with (e.g. there will only be SSC in non-rainy seasons for portion of TP with seasonal cloud cover, ice cover). Was data filtered to sites with representative data first? Was only summer season data used?

Response: We admire your professionalism and scientific rigor. In the chapter on *Data and Quality Control*, we have detailed our selection criteria for satellite products (lines 361-383) and the preliminary quality assurance measures (lines 415-440). In all analyses, we also elaborated on our preprocessing steps, specifically addressing the removal of environmental noise (ice, glacier, snow, and mountain shadows) and the seasonal interferences you mentioned. We have meticulously filtered out pixels flagged as clouds, ice, land, and image artifacts from the water masks we extracted. As you

correctly noted, this filtering process is an integral part of the algorithm, specifically carried out during the initial pre-processing stage, and it extends across the entire dataset, encompassing around 76,000 Landsat images. This comprehensive pre-processing is applied universally, not just to locations with representative data.

For the TP, it's important to acknowledge the occurrence of winter river freezing, particularly in permafrost regions and headwaters around the Himalayas. In these regions, ice and snow pixels have already been effectively excluded in the pre-processing module. However, in the eastern and southern regions of TP (e.g., Yangtze River, Salween River), where rivers remain ice-free throughout the year, SSC observations are ongoing year-round.

Furthermore, during winter when land surfaces and river freeze, processes such as overland flow erosion, active layer thaw-freeze erosion, and riverine sediment transport come to a standstill (Li et al., 2021b; J. Li et al., 2023). In essence, the effectiveness of our algorithm in identifying river pixels (or water masks) serves as evidence that we have successfully excluded the temporal bias and other environmental factors inherent in satellite imagery (see Method section, 6. Analyzing trends in suspended sediment dynamics).

You can verify the spatial variations in our extraction of river pixels; all river pixels exhibit a positive correlation with river width (Fig. S1C). Some spatial variation in the number of samples results from variations in cloud- and snow/ice-free days, as well as image losses due to retrieval and/or storage challenges in the Landsat program (Fig. S1C). Consequently, trends for some rivers, particularly in small rivers, may be incompletely captured by these methods. Nonetheless, the spatial breadth and temporal scope of these data allow for analysis of many rivers without current and/or historical in situ monitoring programs. We have also thoroughly reviewed and expanded on these processing methods for clarity. We have also included 'Limitations and Perspectives' section in the Supplementary Information to discussing this.

To offer a more lucid representation of the performance of the Landsat data we employed throughout the year, we conducted a monthly distribution analysis of the corresponding Landsat images (1986-2021). To provide you and our readers with a more perspicuous illustration of the annual data distribution, we have also developed a presentation code. You can effortlessly review the annual sampling frequency distribution of Landsat 5, 7, and 8 images by following the link provided (Year-Month-Day, where we carried out monthly evaluations). Furthermore, you can pinpoint the sampling locations for the respective images based on the official Landsat satellite path and row numbers (WRS_PATH and WRS_ROW) within the code. We have also introduced a new figure, Supplementary Fig. S18, designed to showcase the monthly distribution of SSC and flux assessed by the satellite throughout the year.

<https://code.earthengine.google.com/0da2ce05dbef69d61fed40eb86c134f1?noload=true>

-The phrase “source to sink processes” is used many times and is a good phrase to frame the paper at the beginning, but I recommend getting more specific later on in the paper as this phrase is too general and can mean many things (e.g. are you talking about bed/bank erosion, deposition, flux, burial etc.)

Response: We appreciate the thorough review by the reviewer 2. The descriptions related to source-to-sink processes, as well as other generalized terms and sentences, have been refined for greater precision. You will clearly notice our meticulous revisions throughout the revised manuscript to meet your linguistic expectations (e.g., lines 33-37, 49-53, 61-66, 112-115, 149-152, 226-248, etc.).

Lines 25-27: Are the % increases in SSC and flux in comparison to “previously reported” studies or over the time period within this study? Please clarify. Also, if referring to previous studies, this is a difficult comparison to make in an abstract considering differences in methods/time periods etc. in previous studies. This result reported in the abstract didn’t seem to be a focus and I couldn’t find it in the manuscript. I would recommend reporting only major results from your study in abstract. What are the main one or two results you really want people to know?

Response: To address the reviewer’s concern, we have comprehensively revised the abstract, as you rightly pointed out, the previous description was disproportionately misaligned with the main theme of our research. We adjusted the content and retained certain expressions, believing they would pique the interest of readers, especially researchers in this field, towards our paper. Furthermore, we delved deeper into outlining the core conclusions of the article and, heeding your advice, endeavored to keep the language succinct. We have also included a new paragraph in the *Results and Discussion* (see lines 233-248) to compare our findings with previous literature. We meticulously controlled and employed the data for comparison, thereby quantitatively revealing the potential underestimation of sediment flux in rivers based on site-specific studies for the first time.

In line with your feedback and that of Reviewer 1, this study predominantly relied on remote sensing imagery to explore spatiotemporal trends in the headwaters of major Asian rivers. In light of these insights, we have restructured the Abstract for the new version, following this sequence: 1) An emphasis on the research background, 2) a clarification of present limitations, 3) an outline of the core contributions of this paper, 4) a revelation of the primary findings, and 5) an elucidation of the broader impact and significance of this work within the current research landscape (See Abstract).

Line 29-31: “Rapid shifts in the erosion-deposition patterns have regulated the sediment source to sink processes” … This sentence is an example of the type of

wording that is frequently used but is too vague to have meaning. While I understand the general idea here, and its good to be general in the abstract, I recommend getting a little more specific as this sentence is difficult to interpret.

Response: We appreciate your pointing this out. We have revised the phrasing of that sentence (in new Abstract) to make it more concise and easily comprehensible to readers, even if they are not involved in research related to this field.

Lines 34-36: Consider revising this opening sentence and wording. 1) I assume “water energy” is referring to hydropower? Please clarify. 2) Hydropower is the only item in this list that is provided to people, while ecological function and sediment/carbon/nutrients are provided to ecosystems not people. Please revise this phrasing. 3) Is 2 billion people the population in the TP? If so please clarify and note TP rivers do not supply these things “worldwide” but only in the TP basin, so remove the word worldwide.

Response: We appreciate the reviewer's insightful recommendation, which underscores the importance of clarity and precision in our descriptions. We agree that terms like 'water resources' and 'ecological functions' should be articulated more precisely (See *Introduction*, lines 40-44 in the first paragraph). Furthermore, the reference to the two billion people pertains to those influenced by rivers originating from the TP, encompassing populations in surrounding and downstream areas—including China, India, parts of Central Asia, and most countries in the Indochina Peninsula. In line with your General Comment #1, we are committed to employing more precise and specific language in our resubmission to make our work more compelling for Nature Communications' readership.

Line 41: What is a sedimentary effect?

Response: ‘Sedimentary effects’ refer to the impacts or consequences related to sediment transport, deposition, and accumulation. In the context of river systems, these effects can include deposition, stratification, compaction, and lithification. However, we agree with the reviewer that we modified the whole text with more precise words, thus we simplify the associated descriptions in the first paragraph. This revision makes the sentence more concise while still conveying the core message.

Line 43: Is sediment availability refer to sediment supply? And availability in the landscape or within the river channel? Please be specific when possible.

Response: Certainly, sediment availability is not synonymous with but encompasses the notion of sediment supply, extending to both the landscape and the river channel (Najafi et al., 2021; Zhang et al., 2022). The effects of changes in erosion and deposition patterns pivot on whether these changes result in an augmentation of sediment availability or not (<https://www.unep.org/cep/sedimentation-and-erosion>). We agree with

your emphasis on precise language, and relevant words have been replaced accordingly in our resubmission.

Line 46-49: changes in sediment delivery in all environments is the balance of mobilization, transport, and deposition complicated by climate change. Consider getting more specific what this means in the context of “alpine cold environments”.

Response: We appreciate your comment on this point. Rather than using abstract or conceptual language, we now provide a detailed, quantitative description of sediment dynamics on the TP. Our resubmission contrasts these findings with sediment erosion studies in pan-Arctic and Greenland regions, emphasizing the pressing nature of TP challenges. We've refined the narrative for logical clarity, and following your suggestions, we've added data-rich statements to clarify our study's context (See *Introduction*, the second paragraph).

Line 50: Note, sediment mobilization and transport are examples of “source-to-sink” processes, consider removing phrase source to sink here and/or be more specific.

Response: In response to your insightful comments on our language, especially within the introduction's initial paragraphs, we have endeavored to use a more accessible yet scientifically precise language. This will better elucidate our study's context for readers. As a result, we've thoroughly refined the introduction and replaced any ambiguous words (e.g., lines 33-37, 49-53, 61-66, 112-115, 149-152, 226-248, etc.).

Line 54: What is mean by “geomorphic feedback on these changes”? Consider being more specific or removing this phrase.

Response: To address the reviewer's comment, we have restructured the second paragraph of the Introduction to (a) provide an overview of sediment changes in the TP. (b) highlight the ecological and geomorphological significance of suspended sediment. (c) introduce a transition emphasizing the current knowledge gaps. We have particularly emphasized how sediment changes trigger geomorphic responses, namely “geomorphic feedback” as you pointed out (See *Introduction*, the second paragraph). We believe that such a detailed portrayal aligns with your expectations for clarity in our language expression.

Line 85: This first paragraph seems to be more about spatial heterogeneity or do some of these statistics and sediment yields also refer to temporal trends? Please clarify and perhaps focus this paragraph on spatial patterns if that is your goal. So remove the word “spatiotemporal” if this is only spatial.

Response: Reviewer 2 astutely identified that our initial paragraph primarily focuses on spatial heterogeneity. It is true. This aligns with the inaugural subsection of our 'Results and Discussion', named 'Spatial heterogeneity in SSC and suspended sediment

flux'. The design of this subsection unfolds as:

(a) Emphasis on spatial differences in sediment production, illustrated by overarching SSC variations and sediment yield divergences in four key basins.

(b) A discourse on the temporal shifts in suspended sediment concentration (SSC, the second paragraph). It's crucial to underscore that for this analysis, SSC serves as a more apt metric than basin sediment yield. This choice stems from the differential responses of various basins to climate change, with SSC offering a straightforward portrayal of temporal sediment dynamics in rivers.

(c) The concluding segments expound on the causes driving this spatiotemporal discrepancy (the third and fourth paragraph).

We've thus refined the section and remove the word “spatiotemporal” to align with its overarching theme and title more fittingly.

Line 137: “Furthermore, the thawing of permafrost has led to the expansion of thermokarst landscapes and hillslope mass wasting, which resulted in a relative higher SSCs where permafrost is developed on the TP” . Is this an interpretation from the SSC data or relationships you found between thawing permafrost and SSC? Please clarify.

Response: We agree with your insight on this matter. Our assertion stems from previous studies that identified a link between permafrost thawing and heightened SSC levels on the TP (Li et al., 2021b; Shi et al., 2022). It's important to note that our interpretation was not directly extrapolated from our SSC data but rather was used to frame our findings in the broader scientific discourse. To enhance clarity and sidestep potential misinterpretations, we've refined this section and embedded relevant citations to more effectively convey the context and our perspective.

Additionally, our multi-year active layer and ground temperature probe data have been cross-referenced with in-situ SSC measurements, further bolstering our stance. These collective observations have been consolidated into a Supplementary Fig. S17, offering both empirical and literature-backed support for our assertions.

Figure 1. Are these data long-term average or annual averages as noted in caption?

Response: Figure 1 showcases annual averages, as indicated in the notes of this figure. We will ensure that the caption is rectified to consistently reflect this and to align with the data presented within the figure. Thank you for highlighting this discrepancy. Additionally, we have meticulously reviewed the entire manuscript to ensure that the statistical methods applied to the data are clearly articulated.

Line 167-170: Consider rephrasing this sentence and be more specific. This is an

example of a general statement found throughout the manuscript that is difficult to interpret because it's too general.

Response: We extend our gratitude once again for your attention to the specific language and phrasing issues in our manuscript (See lines 176-181). All co-authors have meticulously reviewed the entire document, refining the language sentence by sentence. We ensure that our language is detailed and specific, catering to the readership of Nature Communications.

Line 222 and Table S2: Does the underestimation of sediment flux refer to comparison between yields in this study and other studies? If so, where are the sediment yields in Table 2 from other studies? Also if so, I assume other studies were not satellite based but hopefully they are based on surface SSC samples because satellite based SSC and flux will almost always be underestimated compared to a depth-integrated field based measurement. Just making sure your interpretation of underestimated fluxes is not caused by differences in data and satellite biases instead of the deposition processes mentioned.

Response: Your insight on the potential discrepancy between satellite-based SSC and depth-integrated field-based measurements is indeed very appreciated. Satellite-based SSC and flux indeed have inherent limitations, including potential underestimation, due to various reasons such as sub-pixel variability, atmospheric interference, or the satellite's inability to capture depth-varied sediment concentrations. However, we have rigorously calculated the discrepancy between the sediment yield of each relative ratio outlet reported and the sediment yield of the entire river segment as captured by satellite statistics. As indicated in the footnote of Table S2, which we invite you to refer to, we clearly depict the methodology used to estimate and compute these underestimations. Our comparison is rooted in juxtaposing the satellite-derived output values against the corresponding output values from the station data. Additionally, we have meticulously reviewed our data sources and have marked data that were measured by a depth-integrated field-based measurement approach (Supplementary Table S2, subscript annotation with *). This labeling is provided to facilitate better comparison for you.

We have meticulously used depth-integrated SSC samples to train and calibrate our model (lines 349-353). The sediment yields in Table 2, derived from literature, are based on in-situ sediment flux measurements at basin outlets, which are then scaled with the basin area. Consequently, it stands to reason that this value would be less than the sediment yield for the entire basin. The discrepancy is attributed to the hindrance of high SSC discharges within the basin and is not due to biases from satellite measurements or sampling methods. This observation can be discernibly identified in Table S2 where not only concentrations (which are inferred directly from band values) but also fluxes exhibit a degree of underestimation. The disproportionately large, basin-weighted SSC values are evidently due to the non-discharge of high SSC from within the basin downstream. This interpretation is supported by the calibrations we conducted

on the reported SSCs at the outlet stations. To elucidate further, the following formula might assist you in understanding this point more clearly:

$$SSY(\text{Suspended sediment yield}) = \frac{SSF}{A} \quad (1)$$

$$SSF = SSC * Q \quad (2)$$

Where SSF(suspended sediment flux) represents the total sediment discharge, often in units such as tons/year or kg/year. And A stands for the area of the catchment or watershed, typically measured in square kilometers or square meters. SSC, suspended sediment concentration. Q is the discharge.

Additionally, in Table S3, we have meticulously quantified the sediment proportions for river segments, based on their lengths. The determination of these segments was anchored on the computed SSC anomalies (when juxtaposed with multi-year averages). We are confident that this dual-pronged explanation should convince you that the observed discrepancies stem from sedimentation processes.

Some papers that may be useful background or citations:

Balasubramanian S V et al 2020 Robust algorithm for estimating total suspended solids (TSS) in inland and nearshore coastal waters Remote Sens. Environ. 246 111768

Gardner et al 2023. Human activities change suspended sediment concentration along rivers. Environ. Res. Lett. 18 064032

Claverie M, Vermote E F, Franch B and Masek J G 2015 Evaluation of the Landsat-5 TM and Landsat-7 ETM+ surface reflectance products Remote Sens. Environ. 169 390 – 403

Response: We once again express our gratitude to Reviewer 2 for the meticulous scrutiny of our manuscript. The references you pointed out have indeed enriched our discussion, and we have incorporated them appropriately throughout the text.

References

- Ali, K.F., De Boer, D.H., 2007. Spatial patterns and variation of suspended sediment yield in the upper Indus River basin, northern Pakistan. *Journal of Hydrology* 334, 368–387. <https://doi.org/10.1016/j.jhydrol.2006.10.013>
- Balasubramanian, S.V., Pahlevan, N., Smith, B., Binding, C., Schalles, J., Loisel, H., Gurlin, D., Greb, S., Alikas, K., Randla, M., Bunkei, M., Moses, W., Nguyễn, H., Lehmann, M.K., O'Donnell, D., Ondrusek, M., Han, T.-H., Fichot, C.G., Moore, T., Boss, E., 2020. Robust algorithm for estimating total suspended solids (TSS) in inland and nearshore coastal waters. *Remote Sensing of Environment* 246, 111768. <https://doi.org/10.1016/j.rse.2020.111768>
- Claverie, M., Vermote, E.F., Franch, B., Masek, J.G., 2015. Evaluation of the Landsat-5 TM and Landsat-7 ETM+ surface reflectance products. *Remote Sensing of Environment* 169, 390–403. <https://doi.org/10.1016/j.rse.2015.08.030>
- Dethier, E.N., Renshaw, C.E., Magilligan, F.J., 2022. Rapid changes to global river suspended sediment flux by humans. *Science* 376, 1447–1452. <https://doi.org/10.1126/science.abn7980>
- Dethier, E.N., Renshaw, C.E., Magilligan, F.J., 2020. Toward Improved Accuracy of Remote Sensing Approaches for Quantifying Suspended Sediment: Implications for Suspended-Sediment Monitoring. *JGR Earth Surface* 125. <https://doi.org/10.1029/2019JF005033>
- Dethier, E.N., Silman, M., Leiva, J.D., Alqahtani, S., Fernandez, L.E., Pauca, P., Çamalan, S., Tomhave, P., Magilligan, F.J., Renshaw, C.E., Lutz, D.A., 2023. A global rise in alluvial mining increases sediment load in tropical rivers. *Nature* 620, 787–793. <https://doi.org/10.1038/s41586-023-06309-9>
- East, A.E., Sankey, J.B., 2020. Geomorphic and Sedimentary Effects of Modern Climate Change: Current and Anticipated Future Conditions in the Western United States. *Reviews of Geophysics* 58. <https://doi.org/10.1029/2019RG000692>
- Harel, M.-A., Mudd, S.M., Attal, M., 2016. Global analysis of the stream power law parameters based on worldwide ¹⁰Be denudation rates. *Geomorphology* 268, 184–196. <https://doi.org/10.1016/j.geomorph.2016.05.035>
- Islam, Z., 2022. Soil loss assessment by RUSLE in the cloud-based platform (GEE) in Nigeria. *Model. Earth Syst. Environ.* 8, 4579–4591. <https://doi.org/10.1007/s40808-022-01467-7>
- Li, D., Lu, X., Overeem, I., Walling, D.E., Syvitski, J., Kettner, A.J., Bookhagen, B., Zhou, Y., Zhang, T., 2021a. Exceptional increases in fluvial sediment fluxes in a warmer and wetter High Mountain Asia. *Science* 374, 599–603. <https://doi.org/10.1126/science.abi9649>
- Li, D., Overeem, I., Kettner, A.J., Zhou, Y., Lu, X., 2021b. Air Temperature Regulates Erodible Landscape, Water, and Sediment Fluxes in the Permafrost-Dominated Catchment on the Tibetan Plateau. *Water Res.* 57. <https://doi.org/10.1029/2020WR028193>
- Li, J., Wang, G., Li, K., Li, Y., Guo, L., Song, C., 2023. Impacts of climate change and freeze–thaw cycles on water and sediment fluxes in the headwater region of the

- Yangtze River, Qinghai–Tibet Plateau. *CATENA* 227, 107112. <https://doi.org/10.1016/j.catena.2023.107112>
- Li, X., Long, D., Slater, L.J., Moulds, S., Shahid, M., Han, P., Zhao, F., 2023. Soil Moisture to Runoff (SM2R): A Data-Driven Model for Runoff Estimation Across Poorly Gauged Asian Water Towers Based on Soil Moisture Dynamics. *Water Resources Research* 59, e2022WR033597. <https://doi.org/10.1029/2022WR033597>
- Morley, S.K., Brito, T.V., Welling, D.T., 2018. Measures of Model Performance Based On the Log Accuracy Ratio. *Space Weather* 16, 69–88. <https://doi.org/10.1002/2017SW001669>
- Najafi, S., Sadeghi, S.H., Heckmann, T., 2021. Analysis of sediment accessibility and availability concepts based on sediment connectivity throughout a watershed. *Land Degrad Dev* 32, 3023–3044. <https://doi.org/10.1002/ldr.3964>
- Overeem, I., Hudson, B.D., Syvitski, J.P.M., Mikkelsen, A.B., Hasholt, B., van den Broeke, M.R., Noël, B.P.Y., Morlighem, M., 2017. Substantial export of suspended sediment to the global oceans from glacial erosion in Greenland. *Nature Geosci* 10, 859–863. <https://doi.org/10.1038/ngeo3046>
- Scaramuzza, P., Micijevic, E., Chander, G., 2004. SLC gap-filled products: Phase one methodology.
- Shi, X., Zhang, F., Lu, X., Zhang, Y., Zheng, Y., Wang, G., Wang, L., Jagirani, M.D., Wang, T., Piao, S., 2022. The response of the suspended sediment load of the headwaters of the Brahmaputra River to climate change: Quantitative attribution to the effects of hydrological, cryospheric and vegetation controls. *Global and Planetary Change* 210, 103753. <https://doi.org/10.1016/j.gloplacha.2022.103753>
- Syvitski, J., Ángel, J.R., Saito, Y., Overeem, I., Vörösmarty, C.J., Wang, H., Olago, D., 2022. Earth's sediment cycle during the Anthropocene. *Nat Rev Earth Environ* 3, 179–196. <https://doi.org/10.1038/s43017-021-00253-w>
- Teng, H., Liang, Z., Chen, S., Liu, Y., Viscarra Rossel, R.A., Chappell, A., Yu, W., Shi, Z., 2018. Current and future assessments of soil erosion by water on the Tibetan Plateau based on RUSLE and CMIP5 climate models. *Science of The Total Environment* 635, 673–686. <https://doi.org/10.1016/j.scitotenv.2018.04.146>
- Tian, Q., Xu, K.H., Dong, C.M., Yang, S.L., He, Y.J., Shi, B.W., 2021. Declining Sediment Discharge in the Yangtze River From 1956 to 2017: Spatial and Temporal Changes and Their Causes. *Water Res* 57. <https://doi.org/10.1029/2020WR028645>
- Wang, L., Yao, T., Chai, C., Cuo, L., Su, F., Zhang, F., Yao, Z., Zhang, Y., Li, X., Qi, J., Hu, Z., Liu, J., Wang, Y., 2021. TP-River: Monitoring and Quantifying Total River Runoff from the Third Pole. *Bulletin of the American Meteorological Society* 102, E948–E965. <https://doi.org/10.1175/BAMS-D-20-0207.1>
- Wang, S., Fu, B., Piao, S., Lü, Y., Ciais, P., Feng, X., Wang, Y., 2016. Reduced sediment transport in the Yellow River due to anthropogenic changes. *Nature Geosci* 9, 38–41. <https://doi.org/10.1038/ngeo2602>
- Zhang, T., Li, D., East, A.E., Walling, D.E., Lane, S., Overeem, I., Beylich, A.A.,

- Koppes, M., Lu, X., 2022. Warming-driven erosion and sediment transport in cold regions. *Nat Rev Earth Environ* 3, 832–851. <https://doi.org/10.1038/s43017-022-00362-0>
- Zhang, T., Li, D., Kettner, A.J., Zhou, Y., Lu, X., 2021. Constraining Dynamic Sediment-Discharge Relationships in Cold Environments: The Sediment-Availability-Transport (SAT) Model. *Water Resources Research* 57. <https://doi.org/10.1029/2021WR030690>

REVIEWERS' COMMENTS

Reviewer #1 (Remarks to the Author):

I have reviewed this manuscript during the first round. I have now read through the authors response to comments, and I was glad to see their detail response to my earlier comments. I have no further major issues with this study; the manuscript reads well and it will be an important contribution to the geophysical sciences.

Reviewer #2 (Remarks to the Author):

The authors have done an great job revising this manuscript. I have no further comments.

Specific comments:

Line 349: I think there is a small typo: "deep-integrated" should be "depth-integrated", correct?